# Stoichiometric interactions explain spindle dynamics and scaling across 100 million years of nematode evolution

Reza Farhadifar[1,2]*, Che-Hang Yu[1†], Gunar Fabig[3†], Hai-Yin Wu[1], David B Stein[2], Matthew Rockman[4], Thomas Müller-Reichert[3], Michael J Shelley[2,5], Daniel J Needleman[1,2]

[1]Department of Molecular and Cellular Biology and School of Engineering and Applied Sciences, Harvard University, Cambridge, United States; [2]Center for Computational Biology, Flatiron Institute, New York, United States; [3]Experimental Center, Faculty of Medicine Carl Gustav Carus, Dresden, Germany; [4]Department of Biology and Center for Genomics & Systems Biology, New York University, New York, United States; [5]Courant Institute, New York University, New York, United States

**Abstract** The spindle shows remarkable diversity, and changes in an integrated fashion, as cells vary over evolution. Here, we provide a mechanistic explanation for variations in the first mitotic spindle in nematodes. We used a combination of quantitative genetics and biophysics to rule out broad classes of models of the regulation of spindle length and dynamics, and to establish the importance of a balance of cortical pulling forces acting in different directions. These experiments led us to construct a model of cortical pulling forces in which the stoichiometric interactions of microtubules and force generators (each force generator can bind only one microtubule), is key to explaining the dynamics of spindle positioning and elongation, and spindle final length and scaling with cell size. This model accounts for variations in all the spindle traits we studied here, both within species and across nematode species spanning over 100 million years of evolution.

*For correspondence:
rfarhadifar@flatironinstitute.org

†These authors contributed equally to this work

Competing interests: The authors declare that no competing interests exist.

## Introduction

Cell division is a highly complex process requiring the spatial and temporal coordination of many events. Since cells vary over evolution (and through the course of development), the various aspects of the cell division machinery must change in an integrated way to continue to work together in these different contexts. Recently, several groups have investigated variations of this machinery with cell size, finding that spindle size scales with cell size (*Brown et al., 2007*; *Brust-Mascher et al., 2004*; *Decker et al., 2018*; *Good et al., 2013*; *Greenan et al., 2010*; *Hara and Kimura, 2009*; *Hazel et al., 2013*; *Loughlin et al., 2011*; *Reber et al., 2013*; *Wilbur and Heald, 2013*; *Lacroix et al., 2018*; *Rizk et al., 2014*). Many other aspects of cell division also change with cell size, including the dynamics and positioning of the spindle, reorganization of organelles and the cytoplasm, and rearrangements of the cortex that ultimately result in the division of the cell (*Hara and Kimura, 2009*; *Hara and Kimura, 2011*; *Blanchoud et al., 2015*; *Carvalho et al., 2009*). The mechanisms by which these processes change in a coordinated fashion over development and evolution are poorly understood.

The *Caenorhabditis elegans* embryo is a powerful model system that has been extensively used to study cell division and scaling (*Greenan et al., 2010*; *Hara and Kimura, 2009*; *Lacroix et al., 2018*; *Carvalho et al., 2009*; *Wu et al., 2017*; *Weber and Brangwynne, 2015*; *Garzon-Coral et al., 2016*; *Grill et al., 2003*; *Hara and Kimura, 2013*; *Kozlowski et al., 2007*; *Labbé et al., 2004*;

*Ladouceur et al., 2015*; *Pécréaux et al., 2016*; *Pecreaux et al., 2006*; *Redemann et al., 2010*; *Decker et al., 2011*; *Fielmich et al., 2018*). The first cell division in *C. elegans* is asymmetric: the two daughter cells have different sizes and fates. The asymmetry of the single-cell embryo is established shortly after fertilization, with the cell cortex divided into two domains enriched in either anterior or posterior partitioning-defected proteins (PARs) (*Kemphues et al., 1988*). The spindle forms in the center of the embryo, and is a bipolar structure primarily composed of transitory microtubules and their associated proteins, with centrosomes localized at each pole. Centrosomes are organizing centers which nucleate microtubules. Astral microtubules are organized around the centrosomes, radiating away from them towards the cell cortex. Subsequent elongation and asymmetric positioning of the spindle is driven by factors, asymmetrically localized by the PAR proteins, which exert pulling forces on astral microtubules (*Colombo et al., 2003*; *Grill et al., 2001*). The embryo then divides asymmetrically due to the asymmetric positioning of the spindle.

Here, we investigate the regulation and coordination of anaphase spindle elongation and positioning in *C. elegans* single-cell embryos. First, we use quantitative genetics to rule out broad classes of models of spindle size regulation, and to establish the importance of cell length. We also discover two genetic loci that impact spindle length independently of cell length, and argue that these affect cortically localized force generators that pull upon astral microtubules. We next use laser ablation to directly assess the nature of forces acting on the spindle. This shows that spindle motion results from astral microtubules pulling from many directions, with spindle motion ceasing only when those pulling forces are in balance. We constructed a model of cortical pulling forces, based on known biochemical properties of microtubules and molecular motors, and find that it reproduces the dynamics of spindle positioning and elongation, and spindle final size and scaling with cell length. Central to this model are its stoichiometric interactions between microtubules and force generators (each force generator can bind only one microtubule). Stoichiometric interactions lead to a competition of centrosomes for cortical force generators and yield a stable final position. Finally, we show that the Stoichiometric Model accounts for variations in all the spindle traits studied here, across nematode species spanning over 100 million years of evolution.

## Results

### Quantitative perturbations of cell biological phenotypes using natural genetic variation

We exploit the genetic diversity present in a panel of *C. elegans* recombinant inbred advanced intercross lines (RIAILs) to study quantitative variations in spindle length and other related cell biological traits. The founding lines of the RIAILs were the laboratory strain N2 (Bristol) and the Hawaiian natural isolate CB4856, whose genomes differ at approximately one in ~240 base pairs (*Kim et al., 2019*). The panel of 182 RIAILs was generated from ten rounds of random intercrossing, followed by ten rounds of selfing (*Rockman and Kruglyak, 2009*, see *Figure 1—figure supplement 1*), and the final lines were genotyped at 1,454 markers along the genome.

To quantify spindle variation across the RIAIL panel, we developed a high-throughput microscopy platform to image the first mitotic division (*Figure 1—video 1*). We imaged ~50 embryos per line in each of the 182 lines (~five replicates per line, ~10 embryos per replicate, *Figure 1A*) using 3D time-lapse differential interference contrast microscopy (DIC, *Figure 1B*). This resulted in a total of ~12,000,000 microscopy pictures from the 3D movies of the 9,641 embryos that were imaged (*Figure 1C*). We used custom automated image analysis software (*Farhadifar and Needleman, 2014*) to segment and track the spindle and centrosomes during the first cell division in these movies (*Figure 1D*, *Figure 1—video 2*). From this data, we extracted characteristics of the spindle in each embryo, including the initial spindle length, the rate and duration of spindle elongation, and the final spindle length (*Figure 1E*). For each embryo, we also measured a range of other traits, including the position of the centrosomes relative to the cell periphery, the size of the cell, and the position of the division plane (*Figure 1—figure supplement 2*). The founding *C. elegans* strains used to create the RIAILs, N2 and CB4856, had similar spindle characteristics (*Figure 1—figure supplement 3A*; *Farhadifar et al., 2015*). In contrast, we observed extensive quantitative variations in spindle characteristics across the RIAIL panel (*Figure 1* and *Figure 1—figure supplement 3C-E*). To determine if the differences between lines were statistically significant, we measured the mean and

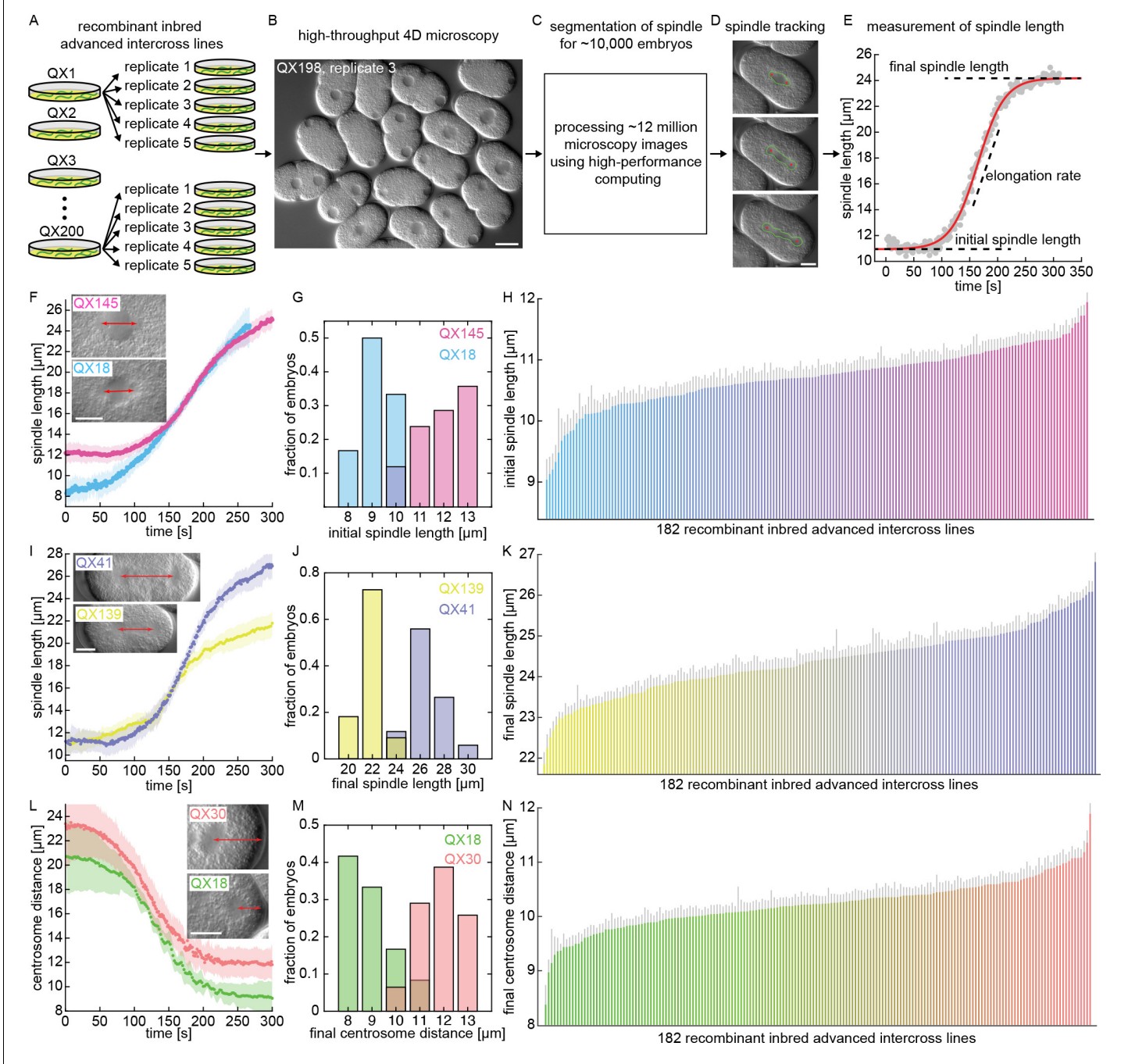

**Figure 1.** High-throughput microscopy and measurements of spindle variation across a panel of *C. elegans* recombinant inbred advanced intercross lines. (A–E) High-throughput microscopy and image processing of *C. elegans* embryos across the RIAIL panel. (A) Five replicates per line, for 182 RIAILs, were imaged. (B) A sample picture from high-throughput 4D DIC microscopy (3D time-lapse) of *C. elegans* embryos. Scale bar 20 μm. (C) Automated analysis of ~12,000,000 microscopy images from ~10,000 *C. elegans* embryos using high-performance computing. (D) Segmentation of the mitotic spindle and tracking of its poles (centrosomes) during the first cell division. Scale bar 10 μm. (E) Spindle length as a function of time for the embryo shown in D (red curve, sigmoid function fit to the data). Initial and final spindle length and elongation rate are indicated. (F–N) Quantitative variation in spindle size and positioning across the RIAIL panel (F, I, and L, solid dots, mean; shaded region, standard deviation; H, K, and N, gray line, standard error): (F) Spindle length as a function of time for two RIAILs, QX18 (n = 12) and QX145 (n = 42). Inset shows sample embryos from these two lines (red arrow, distance between spindle poles). Scale bar 10 μm. (G) Measured distribution of the initial spindle length in embryos from QX18 (n = 12) and QX145 (n = 42). (H) Ranked order plot of the line-averaged initial spindle length for each of the 182 RIAILs. (I) Spindle length as a function of time for two RIAILs, QX41 (n = 34) and QX139 (n = 11). Inset shows sample embryos from these two lines (red arrow, distance between spindle poles). Scale bar 10 μm. (J) Measured distribution of the final spindle length in embryos from QX41 (n = 34) and QX139 (n = 11). (K) Ranked order plot of the line-

*Figure 1 continued on next page*

*Figure 1 continued*

averaged final spindle length for each of the 182 RIAILs. (L) Centrosome distance as a function of time for two RIAILs, QX18 (n = 12) and QX30 (n = 31). Inset shows sample embryos from these two lines (red arrow, distance between spindle pole and cell periphery). Scale bar 10 μm. (M) Measured distribution of the final centrosome distance in embryos from QX18 (n = 12) and QX30 (n = 31). (N) Ranked order plot of the line-averaged final centrosome distance for each of the 182 RIAILs.

The online version of this article includes the following video and figure supplement(s) for figure 1:

**Figure supplement 1.** Breeding design for recombinant inbred advanced intercross lines.

**Figure supplement 2.** Cell division traits.

**Figure supplement 3.** Variation of spindle traits across the RIAIL panel.

**Figure 1—video 1.** High-throughput 3D microscopy of spindle in *C. elegans* embryos.

https://elifesciences.org/articles/55877#fig1video1

**Figure 1—video 2.** Segmentation of the spindle in *C. elegans*.

https://elifesciences.org/articles/55877#fig1video2

standard error for spindle traits in each line. We found statistically significant broad-sense heritability (the fraction of variance due to differences between lines) for all spindle traits (*Figure 1—figure supplement 3B*; *Lynch and Walsh, 1998*). Thus, there are genetic variations for the spindle and other cell biological traits across the RIAIL panel, and the phenotypic similarity of these traits in the founding lines results from them having a different genetic basis in those two lines.

## Testing models of spindle size control using variations across recombinant inbred lines

The quantitative variation across the RIAILs provides a tool to test models of spindle size control by using the pattern of variations and co-variations of cell biological traits. We first considered 'Timer' models (*Figure 2A*), which have been proposed for embryonic spindle length control in *Drosophila* (*Brust-Mascher et al., 2004*; *Wollman et al., 2008*). It has been proposed that cortical forces in one-cell *C. elegans* embryos are regulated via a 'Timer' model (*Pecreaux et al., 2006*; *Bouvrais et al., 2018*). In its general form, the Timer model postulates that one set of genetic mechanisms determines the initial spindle length, *IL*, and another independent set of genetic mechanisms regulates the duration and speed at which the spindle elongates. The duration and speed of spindle elongation determine $\Delta$, the extent of spindle elongation during anaphase. These two factors (the initial spindle length and the extent of spindle elongation) determine the final spindle length: $FL = IL + \Delta$ (*Figure 2A*, lower left panel). As the initial spindle length and the extent of elongation of the spindle are postulated to arise from different genetic mechanisms in the Timer model, this model predicts that those traits should vary independently across the RIAILs, and thus that there should be a positive correlation between the initial and final spindle length (*Figure 2A*, lower right panel). To test this prediction, we plotted the average final spindle length vs the average initial spindle length for each of the RIAILs and instead observed a negligible correlation between these two (*Figure 2B*, p = 0.17). Thus, a Timer model does not explain final spindle length in *C. elegans*. The absence of a correlation between initial and final spindle length also argues against models in which the position of centrosomes in metaphase determine the distribution of forces that control the final length of the spindle (*Hara and Kimura, 2009*).

Another class of models, proposed to explain the size regulation of spindles and other organelles, are 'Limiting Component' models (*Good et al., 2013*; *Hazel et al., 2013*; *Decker et al., 2011*; *Goehring and Hyman, 2012*; *Chan and Marshall, 2012*). In this context, these models posit that the spindle elongates until it exhausts the supply of a limiting component, such as tubulin, which is present in the cytoplasm (*Figure 2C*). The amount of a limiting component in a cell is determined by the concentration of that component and the volume of the cell. If it is postulated that independent genetic mechanisms set the concentration of the limiting component and the volume of the cell, then such a model predicts that final spindle length should be positively correlated with those two factors (*Figure 2C*, lower right panel). To test this possibility, we plotted final spindle length as a function of the area of the cell measured from DIC images, which is a proxy for cell volume because of the rotational symmetry of the embryo, across the RIAIL panel, and observed a highly significant correlation (*Figure 2D*, p = 2.97E-36). This result is consistent with this Limiting Component model.

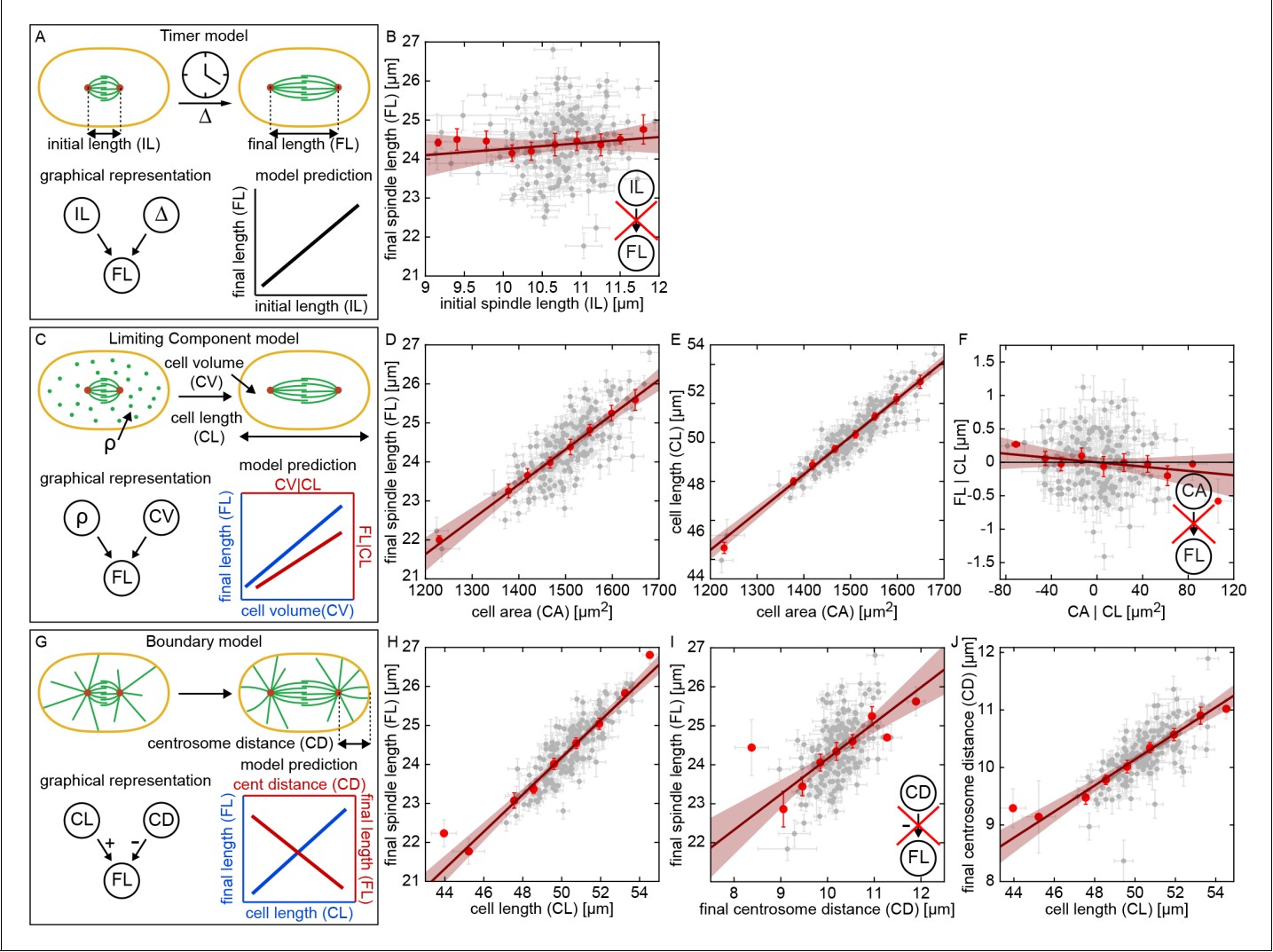

**Figure 2.** Testing models of spindle size control and coordination. (A, C, and G) possible models of spindle size control and associate predictions. (B, D-F) and (H-J) measured correlations and partial correlations of spindle traits to test models (gray dots, mean and standard error of RIAILs; red dots, binned averages; red line, linear fit with 95% prediction bounds). (A) Overview of the Timer model. Initial spindle length (*IL*), spindle elongation (*Δ*), and final spindle length (*FL*) are indicated (*FL = IL + Δ*). (B) Final spindle length as a function of initial spindle length across the RIAIL panel. Inset indicates a lack of correlation between initial and final spindle length, in disagreement with the prediction of the Timer model. (C) Overview of the Limiting Component model. Density of the limiting component (*ρ*), cell volume (*CV*), and cell length (*CL*) are indicated. (D) Final spindle length as a function of cell area across the RIAIL panel. (E) Cell length as a function of cell area across the RIAIL panel. (F) Final spindle length conditioned on cell length (*FL|CL*) as a function of cell area conditioned on cell length (*CA|CL*) across the RIAIL panel. Inset indicates a lack of correlation between cell area and final spindle length conditioned on cell length, in disagreement with the prediction of the Limiting Component model. (G) Overview of the Boundary model. Centrosome distance (*CD*) is indicated. (H) Final spindle length as a function of cell length across the RIAIL panel. (I) Final spindle length as a function of final centrosome distance across the RIAIL panel. Inset indicates a positive correlation between final centrosome distance and final spindle length conditioned, in disagreement with the prediction of the Boundary model. (J) Final centrosome distance as a function of cell length across the RIAIL panel.

However, cell area is highly correlated with cell length across the RIAILs (*Figure 2E*, p = 8.38E-73), making it unclear if cell volume or cell length (or both) contribute to final spindle length. To distinguish between these scenarios, we measured the extent that final spindle length is associated with cell area, independent of cell length. One way to do this would be to measure the correlation between final spindle length and cell volume among RIAILs with the same cell length. A more robust approach is to account for the differences in cell length among the RIAILs by measuring the partial correlation between final spindle length and cell volume: regress final spindle length and cell area

on cell length, and measure the correlation in the residuals (see Materials and methods; for detailed discussion of correlations and partial correlations, see *Kline, 2016*; *Pearl, 2000*). Doing this, we observed a negligible partial correlation between the final spindle length and cell area conditioned on cell length across the RIAILs (*Figure 2F*, p = 0.13). Thus, the correlation between cell area and final spindle length (*Figure 2D*) is due to the correlation between cell area and cell length (*Figure 2E*), and there is no association between cell area and final spindle length that occurs independently of cell length (*Figure 2F*). Therefore, cell volume does not determine final spindle length, which is inconsistent with the simplest version of the Limiting Component model considered here.

Our analysis so far indicates that cell length is a critical factor in the regulation of final spindle length in *C. elegans*. 'Boundary' models are the simplest class of models which have been proposed for size regulation of cellular structures based on cell length. In our context, a Boundary model postulates that the spindle elongates until the centrosomes reach a fixed distance from the cell boundary, perhaps due to a balance of pushing and pulling forces on astral microtubules (MTs). In the simplest version of such a Boundary model, the cell length and the final distance of centrosomes from the cell periphery are determined by independent genetic mechanisms (*Figure 2G*). This model then predicts a positive correlation between final spindle length and cell length (*Figure 2G*, lower right panel, blue) and a negative correlation between final spindle length and centrosome distance (*Figure 2G*, lower right panel, red), that is centrosomes that approach closer to the cell periphery produce longer spindles. To test these predictions, we plot the final spindle length as a function of cell length across the RIAIL panel and find that they are highly correlated as predicted (*Figure 2H*, p = 7.96E-55). However, final spindle length was also positively correlated with centrosome distance (*Figure 2I*, p = 6.81E-14), which is inconsistent with this simplest form of the Boundary model (*Figure 2I*, inset). This inconsistency results from centrosome distance being positively correlated with cell length (*Figure 2J*, p = 5.25E-31), while this Boundary model assumes that they are independent of each other. Thus, the simplest Boundary model does not explain the regulation of final spindle length in *C. elegans*.

## QTL mapping of the genetic basis of spindle size

To gain further insight into the mechanisms that control final spindle length, we next used the RIAIL panel to investigate its genetic basis. The founding lines of the RIAILs, N2 and CB4856, have been sequenced (*Cook et al., 2017*) and each RIAIL was genotyped at 1454 markers along the genome (*Rockman and Kruglyak, 2009*). By comparing the genetic markers and measured traits across the RIAILs, it is possible to identify regions in the genome associated with variations in those traits (*Broman and Sen, 2009*). Such genomic regions, referred to as quantitative trait loci (QTL), contain variants that underlie the genetic basis of those traits. We performed such QTL mapping and discovered multiple QTLs for final spindle length and cell length (*Figure 3—figure supplement 1*). Many QTLs for final spindle length appear to be shared with cell length (*Figure 3—figure supplement 1*), suggesting that those QTLs influence final spindle length by modifying cell length. We next sought to investigate factors that control final spindle length independently of cell length. Hypothetically, this could be done by QTL mapping of final spindle length in subgroups of RIAILs with the same cell length. A more robust approach is to use linear regression to perform QTL mapping of final spindle length conditioned on cell length. This conditional QTL mapping revealed one QTL on chromosome III, which we refer to as QTL1 (*Figure 3A*). We next looked for additional QTLs that might influence final spindle length independent of both cell length and QTL1. Similar to above, we conditioned final length on both cell length and QTL1, which revealed an additional QTL on chromosome III that we refer to as QTL2 (*Figure 3B*). We checked for additional QTLs by conditioning on cell length, QTL1, and QTL2, and did not find any other statistically significant QTLs (*Figure 3—figure supplement 1C*). QTL1 explains 21% of the variation across the RIAILs (i.e. heritability) in final spindle length that is independent of the cell length, while QTL2 explains an additional 10%. Lines with the N2 allele of QTL1 have shorter final spindles (*Figure 3C*), while lines with the N2 allele of QTL2 have longer final spindles (*Figure 3D*). Thus, the N2 alleles of QTL1 and QTL2 have opposite effects on final spindle length (as do the CB4856 alleles). This is consistent with the observation that final spindle length is the same in N2 and CB4856 despite their different genetic basis for this trait as revealed by the variation across the RIAILs.

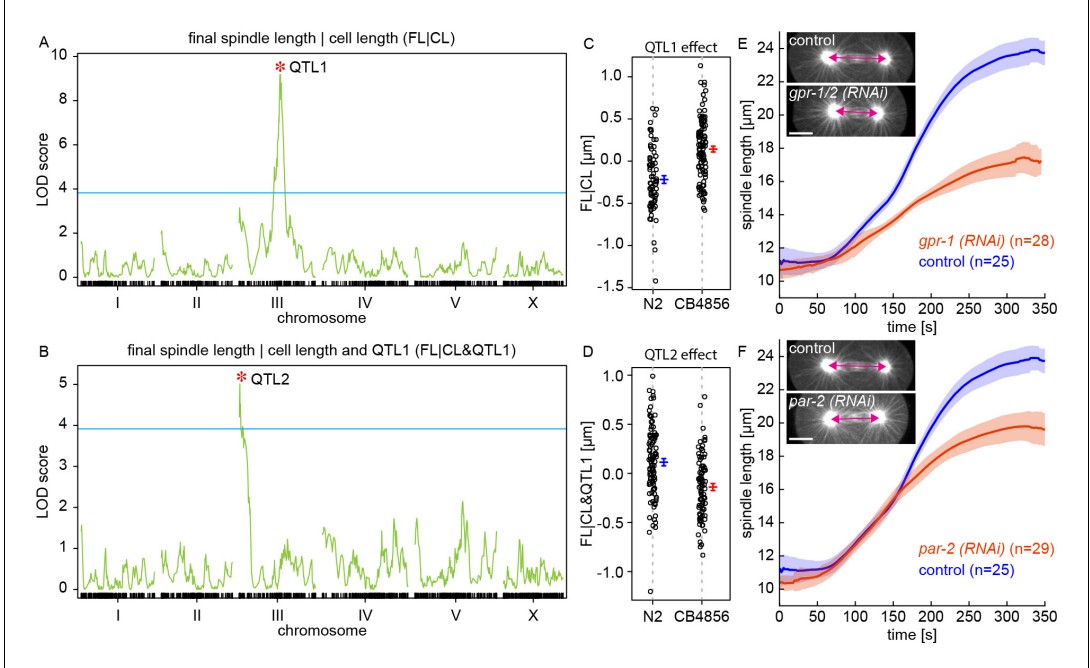

**Figure 3.** QTL mapping of final spindle length. (**A**) QTL mapping of final spindle length conditioned on cell length (*FL|CL*) and (**B**) QTL mapping of final spindle length conditioned on cell length and QTL1 (*FL|CL* and QTL1) (green line, logarithm of the odds (LOD) score; blue line, permutation-based threshold for genome-wide significance at p=0.05; red star indicates the position of a QTL) (**C**) Final spindle length conditioned on cell length for RIAILs grouped by presence of QTL1 variants (blue, N2 variant, mean and standard error; red, CB4856 variant, mean and standard error). (**D**) Final spindle length conditioned on cell length and QTL1 for RIAILs grouped by presence of QTL2 variants (blue, N2 variant, mean and standard error; red, CB4856 variant, mean and standard error). (**E**) Spindle length as a function of time for control (blue) and *gpr-1/2 (RNAi)* (red) embryos (solid line, mean; shaded region, standard deviation). Sample embryos from control and *gpr-1/2 (RNAi)* are shown as insets. Scale bar 10 µm. (**F**) Spindle length as a function of time for control (blue) and *par-2 (RNAi)* (red) embryos (solid line, mean; shaded region, standard deviation). Sample embryos from control and *par-2 (RNAi)* are shown as insets. Scale bar 10 µm.

The online version of this article includes the following video and figure supplement(s) for figure 3:

**Figure supplement 1.** QTL mapping of final spindle length and cell length.

**Figure supplement 2.** Variations of *gpr-1* and *par-2* between N2 and CB4856.

**Figure supplement 3.** Transcript abundance of PAR-2 and GPR1/2.

**Figure 3—video 1.** Spindle elongation in a *gpr-1/2 (RNAi)* embryo.

https://elifesciences.org/articles/55877#fig3video1

**Figure 3—video 2.** Spindle elongation in a *par-2 (RNAi)* embryo.

https://elifesciences.org/articles/55877#fig3video2

QTL1 overlapped with the location of *gpr-1*, which contains two missense variants (nonsynonymous) that differ between N2 and CB4856 (*Figure 3—figure supplement 2*). GPR-1 regulates pulling forces on astral MTs that drive spindle oscillations, positioning, and elongation (*Hara and Kimura, 2009*; *Pecreaux et al., 2006*; *Colombo et al., 2003*). Consistent with previous results (*Hara and Kimura, 2009*), we observed that *gpr-1/2 (RNAi)* embryos have shorter final spindle length (FL = 17.4±0.2 [µm], p = 3.82E-30, *Figure 3E*, *Figure 3—video 1*). QTL2 overlapped with the location of *par-2*, one of the central components of the PAR system in *C. elegans* (*Cuenca et al., 2003*), which contains three missense (nonsynonymous) and two synonymous variants between N2 and CB4856 (*Figure 3—figure supplement 2*). We analyzed previously obtained data from the same panel of RIAILs (*Rockman et al., 2010*) and found that lines with the N2 allele of QTL2 have significantly higher PAR-2 transcript abundance than lines with the CB4856 allele (p = 6.08E-17, *Figure 3—figure supplement 3*; a similar analysis for QTL1 showed no association with GPR-1 transcript abundance, p = 0.22). *par-2 (RNAi)* embryos have shorter final spindle length (FL = 19.8±0.2 [µm], p = 1.03E-22, *Figure 3F*, *Figure 3—video 2*). Previous studies have shown that the PAR system controls the spatial distribution of GPR-1 in *C. elegans* embryos (*Colombo et al., 2003*). While

prior work has described the role of PAR-2 in regulating spindle positioning (*Grill et al., 2001*), this RNAi knockdown shows that PAR-2 also regulates final spindle length.

Taken together, our analysis of the RIAILs shows that cell length is a primary determinant of final spindle length, and identified two QTLs that influence final spindle length independent of cell length. Our RNAi knockdown experiments suggest that the causative genetic variants that underly these QTLs might be in *gpr-1* and *par-2*. The PAR system regulates GPR-1, which is responsible for pulling forces on astral MTs emanating from centrosomes and affects final spindle length. This argues that the spindle elongates until it reaches a length at which pulling forces, and other forces that may be acting on centrosomes, are in balance.

## Dissecting forces on centrosomes using laser ablation

We next characterized the physical forces acting on centrosomes when the spindle has reached its final length. These forces are applied through MTs but it is unclear which MT populations are important for positioning centrosomes. Thus, we used a custom-built laser ablation system to selectively sever different populations of MTs surrounding centrosomes after their motion has ceased and the spindle attained its final length, thereby testing the contribution of those MTs to the balance of forces acting on centrosomes at that time. This system uses kilohertz femtosecond laser pulses to rapidly perform cuts in nearly arbitrary three-dimensional patterns, with minimal collateral damage outside of the ablated region (Materials and methods). While a number of previous studies have used laser ablation to investigate the forces acting on the *C. elegans* mitotic spindle during elongation (*Grill et al., 2003*; *Labbé et al., 2004*; *Krueger et al., 2010*; *Yu et al., 2019*), we are unaware of prior work that used this approach to probe the forces acting on centrosomes after the spindle obtained its final length, as we do here.

We first investigated the hypothesis that the final position of the centrosomes is set by a balance of forces acting on different MTs on the same side of the centrosome (*Garzon-Coral et al., 2016*; *Pécréaux et al., 2016*; *Howard, 2006*; *Zhu et al., 2010*; *Laan et al., 2012*; *Pavin et al., 2012*; *Letort et al., 2016*): with some MTs being subjected to pulling forces from cortical force-generators and other MTs subjected to pushing forces, possibly due to MTs growing against the cell periphery (*Figure 4A*, upper panels). This model predicts that the MTs between the centrosome and the cell periphery are sufficient to maintain their final separation distance, so, if this model is correct, severing MTs emanating from the centrosome in other directions should not impact that distance (*Figure 4A*, lower panels). To test this prediction, we waited until the end of anaphase, when the spindle had finished elongating, and used our laser ablation system to cut a cup like pattern around the centrosome (an elliptical cylinder, open at one end, with minor axis half-length 4.5 µm in *y*, major axis half-length 6 µm in *z*, and a cylinder length of 6 µm) leaving only a cone of MTs extending to the cell periphery, parallel to the spindle axis. Immediately after this cup-cut, the centrosome moved closer to the cell periphery, in the direction of the remaining MTs (*Figure 4B*, *Figure 4—video 1*). We performed this pattern of ablation on 13 embryos (*Figure 4C*) and found that the average speed of centrosome motion after the cut was 14.5±2.8 µm/min, and that the centrosomes moved an average of 3.9±0.2 µm toward the cell periphery. At later times, the centrosomes returned to their original positions from before the cut (*Figure 4—figure supplement 1A, B*), presumably because new MTs grew back and replaced those that were severed. Similar results held for cup-cuts of the anterior centrosome (*Figure 4—figure supplement 1C, D*). The rapid motion of the centrosome in the direction of the remaining MTs after the cup-cut indicates that there is not a balance of pushing and pulling forces acting on these MTs, rather, they are subject to net pulling forces (*Figure 4D*).

Since the centrosome is stationary after spindle elongation, there must be zero net force acting on it. Thus, the pulling forces that astral MTs on one side of the centrosome exert must be balanced by other forces. One hypothesis is that the spindle itself acts as a spring (*Dumont and Mitchison, 2009*; *Goshima and Scholey, 2010*; *Figure 4E*, upper panel). This model predicts that severing the spindle will lead to an imbalance, resulting in pulling forces from astral MTs moving the centrosome closer to the cell periphery (*Figure 4E*, lower panels). To test this prediction, we waited until the end of anaphase, when the spindle had finished elongating, and used our laser ablation system to cut a 4×3 µm plane through the spindle (*Figure 4F*, *Figure 4—video 2*). We performed this pattern of ablation on 15 embryos (*Figure 4G*) and found that the average speed of centrosome motion after the cut was 1.6±1.0 µm/min, and that the centrosomes moved an average of 0.7±0.1 µm toward the

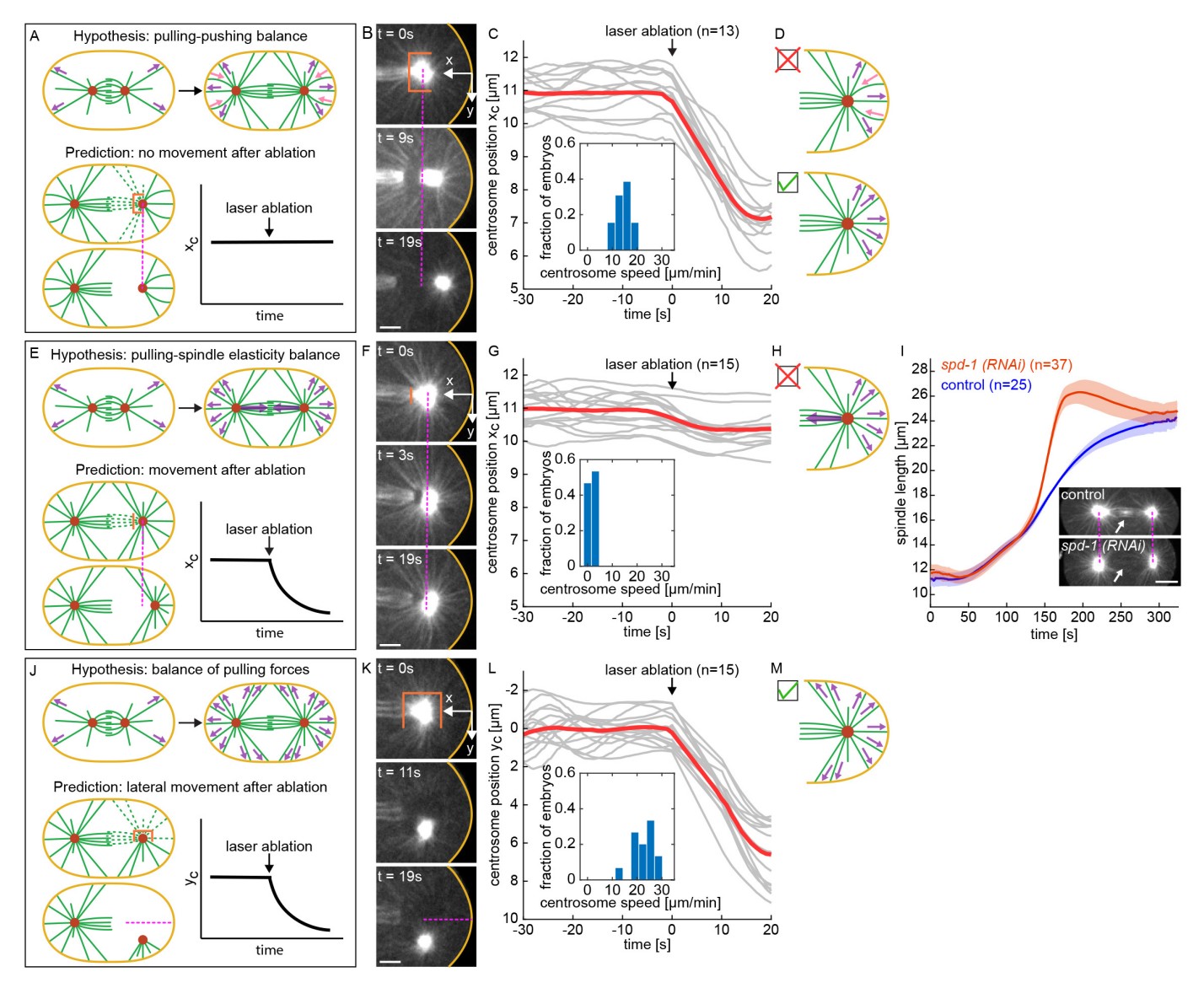

**Figure 4.** Investigating forces on spindle by laser ablation of microtubules. (**A**) Overview of the hypothesis of balanced pulling and pushing (purple arrows, hypothesized pulling forces; pink arrows, hypothesized pushing forces; orange line, ablation geometry; $X_C$, centrosome position in $x$). (**B**) Laser ablation of MTs in a cylindrical geometry with one open end around the centrosome, performed after spindle elongation has ceased. Scale bar 5 μm. (**C**) $x$-position of centrosomes relative to the cell periphery as a function of time for the laser ablation geometry shown in B (gray lines, individual experiments; red line, average). Inset is the histogram of centrosome's speed after ablation. (**D**) Observed centrosome motion after ablation is inconsistent with predictions of balanced pulling and pushing (red cross) and suggests net pulling forces (green check). (**E**) Overview of the hypothesis of balanced pulling and spindle elasticity (purple arrows, hypothesized pulling forces; orange line, ablation geometry; $X_C$, centrosome position). (**F**) Laser ablation of MTs in a planar geometry, performed after spindle elongation has ceased. Scale bar 5 μm. (**G**) $x$-position of centrosomes relative to the cell periphery as a function of time for the laser ablation geometry shown in F (gray lines, individual experiments; red line, average). Inset is the histogram of centrosomes speed after ablation. (**H**) Observed centrosome motion after ablation is inconsistent with predictions of balanced pulling and spindle elasticity (red cross). (**I**) Spindle length as a function of time (solid line, mean; shaded region, standard deviation) for control (blue) and *spd-1 (RNAi)* (red) embryos. Insets are examples of control and *spd-1 (RNAi)* embryos. Dashed lines indicate the position of centrosomes in the control embryo. Scale bar 10 μm. (**J**) Overview of the hypothesis of balanced pulling forces (purple arrows, hypothesized pulling forces; orange line, ablation geometry; $Y_C$, centrosome position). (**K**) Laser ablation of MTs in a cylindrical geometry with one open end around the centrosome, performed after spindle elongation has ceased. Scale bar 5 μm. (**L**) $y$-position of centrosomes relative to the cell periphery as a function of time for the laser ablation geometry shown in K (gray lines, individual experiments; red line, average). Inset is the histogram of centrosomes speed after ablation. (**M**) Observed centrosome motion after ablation is consistent with predictions of balanced pulling forces (green check).

The online version of this article includes the following video and figure supplement(s) for figure 4:

*Figure 4 continued on next page*

**Figure supplement 1.** Centrosome movement after horizontal ablation.

**Figure supplement 2.** Centrosome movement after planar ablation.

**Figure supplement 3.** Centrosome movement after lateral ablation.

**Figure 4—video 1.** Longitudinal cup-cut ablation of MTs.

cell periphery. This result argues that forces from the spindle have a relatively minor impact on the final position of the centrosomes in *C. elegans* (*Figure 4H*). Similar results held for plane-cuts near the anterior centrosome (*Figure 4—figure supplement 2*). To further test this conclusion, we studied *spd-1 (RNAi)* embryos, which lack a central spindle in anaphase (*Figure 4I*, inset) (*Verbrugghe and White, 2004*), and found that their final spindle length is nearly identical to controls (24.5±0.1 µm for *spd-1 (RNAi)* vs 23.8±0.2 µm for control, p = 2.2E-3) (*Figure 4I*, *Figure 4—video 3*). Thus, the central spindle has only a very minor impact on final spindle length. This further argues that the final position of the centrosomes is not set by forces from spindle MTs balancing pulling forces from astral MTs (*Figure 4H*).

Another possibility is that the final position of the centrosomes is set by a balance of cortical pulling forces acting on astral MTs at different orientations (*Figure 4J*, upper panel). In this model, astral MTs at all orientations around the centrosome are subject to cortical pulling forces. Thus, this model predicts that cup-cuts performed at different orientations will cause the centrosomes to move in the direction of the remaining MTs (*Figure 4J*, lower panels). To test this prediction, we waited until the end of anaphase, when the spindle had finished elongating, and used our laser ablation system to perform a cup-cut with the open-end perpendicular to the spindle axis, leaving only a cone of MTs between the centrosome and the cell periphery. Immediately after this cup-cut, the centrosome moved closer to the cell periphery, in the direction of the remaining MTs (*Figure 4K*, *Figure 4—video 4*). We performed this pattern of ablation on 15 embryos (*Figure 4L*) and found that the average speed of centrosome motion after the cut was 22.3±4.3 µm/min, and that the centrosomes moved an average of 6.6±0.3 µm toward the cell periphery. At later times, the centrosomes returned to the original position they were at before the cut (*Figure 4—figure supplement 3A, B*), presumably because new MTs grew back and replaced those that were severed. Similar results held for perpendicular cup-cuts on the anterior centrosome (*Figure 4—figure supplement 3C, D*). Thus, even when centrosome motion has ceased, and the spindle has obtained its final length at the end of anaphase, there are pulling forces present both parallel (*Figure 4B*) and perpendicular (*Figure 4K*) to the spindle axis. This observation is consistent with the hypothesis that the final length of the spindle is determined by a balance of pulling forces acting in different directions on centrosomes (*Figure 4M*).

## The Stoichiometric Model of cortical pulling forces

Our laser ablation experiments argue that the final position of centrosomes after spindle elongation results from a balance of pulling forces acting along different directions. While multiple models of centrosome positioning with cortical pulling forces have been proposed (*Garzon-Coral et al., 2016*; *Grill et al., 2003*; *Pécréaux et al., 2016*; *Laan et al., 2012*; *Pavin et al., 2012*; *Grill and Hyman, 2005*; *Ma et al., 2014*; *Hamaguchi and Hiramoto, 1986*; *Hara and Kimura, 2009*; *Tanimoto et al., 2016*; *Tanimoto et al., 2018*), it is still unclear if cortical pulling forces alone are sufficient to stably position centrosomes (*Howard, 2006*; *Howard and Garzon-Coral, 2017*). To further investigate this issue, we developed a mathematical model of the forces acting on centrosomes due to cortically anchored force-generators (CFGs) pulling on astral MTs (*Figure 5A and B*). In this model, MTs nucleate from centrosomes at rate $\gamma$, grow with velocity $V_g$, and undergo catastrophe at rate $\lambda$. If a MT contacts an unoccupied CFG, it binds and is pulled upon with force $f_0$ along the direction of the MT

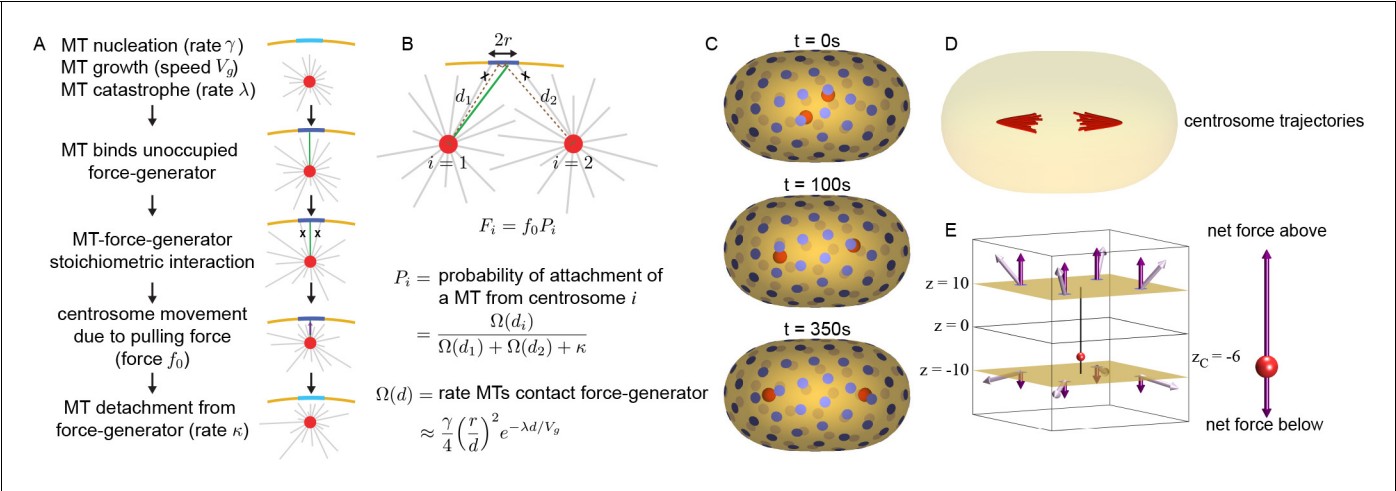

**Figure 5.** Stoichiometric Model of centrosome positioning by cortical pulling forces. (A) From top to bottom: MTs nucleate from the centrosome with rate $\gamma$, grow with speed $V_g$, and undergo catastrophe with rate $\lambda$. An MT (green) that contacts an unoccupied CFG (light blue) becomes bound. Additional MTs (crossed) that contact an occupied CFG (dark blue) do not bind because of their stoichiometric interaction. Bound MTs are pulled with force $f_0$ along the MT direction, causing motion of the centrosome. MTs detach from CFGs with rate $\kappa$. (B) An CFG, with capture radius $r$, in the presence of two centrosomes ($i=1$ and $i=2$) at distances $d_1$ and $d_2$ away. Because of the stoichiometric MT-CFG interactions, only one MT (green) can bind the CFG at a time, so additional MTs (crossed) do not bind the CFG if it is occupied. The average force from the CFG on each centrosome is $F_i = f_0 P_i$, where $P_i$ is the probability of attachment of a MT from centrosome $i$ to the CFG. The probability of attachment, $P_i$, depends on the rate growing MTs impinge upon the CFG, $\Omega(d_i)$, and the MTs detachment rate, $\kappa$. (C) Three-dimensional simulation of two centrosomes (red spheres) in the presence of multiple CFGs (blue disks). (D) Multiple simulations showing centrosome trajectories (red lines) from different initial starting positions close to the cell center. Centrosomes migrate to the same final position irrespective of their initial positions. (E) Simulation of centrosome positioning between two parallel planes with eight CFGs evenly positioned (four on each plane). The pulling force (light purple) and its z-projection (dark purple) for each CFG is shown. The net pulling force on centrosome is shown on the right panel.

The online version of this article includes the following figure supplement(s) for figure 5:

**Figure supplement 1.** Simulations for various parameters.
**Figure supplement 2.** Simulation of spindle elongation for simultaneous varying parameters.
**Figure supplement 3.** Non-stoichiometric model of centrosome positioning by cortical pulling forces.
**Figure supplement 4.** Probability of attachment of microtubules to force-generators.
**Figure supplement 5.** Schematic of centrosome and force-generator geometry.

(*Figure 5A*). Due to the stoichiometric nature of the interaction between molecular motors and MTs, only one MT can bind a CFG at a time. Bound MTs detach from CFGs with rate $\kappa$, leaving the CFG unoccupied (*Figure 5A*). We are primarily interested in the final position of the centrosomes, where they stop moving, so we considered a regime in which the speed of centrosomes is slower than the polymerization and binding dynamics of MTs. In this limit, we calculate $F$, the average pulling force a CFG exerts on a centrosome, which changes over time and depends on the position of the centrosomes. The magnitude of this average pulling force is $P$, the probability of attachment of a MT to the CFG, times $f_0$, the pulling force acting on an attached MT (i.e. $F = f_0 P$). The average pulling force is a function of $d$, the distance between the centrosome and the CFG, because the probability that a MT contacts the CFG is a function of distance. For a single centrosome, we derive the probability of attachment to be:

$$P = \frac{\Omega(d)}{\Omega(d) + \kappa}$$

where $\Omega(d)$, the rate that growing MTs impinge on the CFG, is given by:

$$\Omega(d) \approx \frac{\gamma}{4}\left(\frac{r}{d}\right)^2 e^{-\lambda d/V_g}$$

Here, $r$ is the effective interaction radius of the CFG, which accounts for both the physical size of the CFG and the distance a MT grows and moves along the cortex (Materials and methods).

Because of the stoichiometric nature of the MT-CFG interaction (i.e. only one MT can bind to an CFG at a time), the average force that an CFG exerts on one centrosome is modified by the presence of a second centrosome. This occurs because if a MT from one centrosome binds to an CFG, then MTs from the second centrosome are temporarily prevented from binding to that CFG. The average force on a centrosome at a distance $d_1$ from the CFG, in the presence of a second centrosome at a distance $d_2$ from this CFG, is $F_1 = f_0 P_1$, with $P_1 = \Omega(d_1)/(\Omega(d_1) + \Omega(d_2) + \kappa)$. An analogous expression holds for the average force that this CFG exerts on the second centrosome (*Figure 5B*; Materials and methods).

Using this model, we simulated the motion of two centrosomes inside a cell with a geometry similar to *C. elegans* embryos (cell size = $50 \times 30 \times 30\ \mu m$) with ~100 CFGs evenly distributed on the surface, with parameters estimated from previous experiments (Materials and methods). Centrosomes initially located near the cell center, separate, and move to a final position along the long axis of the cell, at a finite distance from the cell surface (*Figure 5C*). We repeated this simulation with centrosomes starting from different positions and found that centrosomes always move to the same final positions (*Figure 5D*). Changing parameters affects the dynamics of the centrosomes' motion and their final position but gives qualitatively similar results (*Figure 5—figure supplement 1*). Thus, in the Stoichiometric Model, a balance of pulling forces stably positions centrosomes.

Previously, it has been proposed that cortical pulling forces stably position centrosomes if the number of CFGs is less than the number of MTs (*Grill and Hyman, 2005*). To test if that mechanism explains stable centrosome positioning in the Stoichiometric Model, we performed simulations with 10,000 MTs and various number of CFGs (*Figure 5—figure supplement 2*). The centrosomes stably positioned irrespective of the number of CFGs – even for simulations with 100,000 CFGs, ten times more CFGs than MTs (the largest number of CFGs we investigated, Materials and methods). Thus, a limited number of CFGs does not explain the stable positioning of centrosomes in our model. Note that in this model, increasing the number of CFGs does impact the speed of elongation (*Figure 5—figure supplement 1C*). If the force per CFG is correspondingly reduced as the number of CFGs increases, then the dynamics of elongation remain unaltered (*Figure 5—figure supplement 2C, D*). We next investigated the importance of the stoichiometric interaction of MTs and CFGs, that is that CFGs can only bind to one MT at a time, by performing the same simulations without stoichiometric interactions. In this alternative model, all MTs that contact CFGs bind to them and experience a pulling force. In these simulations, the centrosome's positions are always unstable, and the centrosomes migrate until they contact the cell periphery (*Figure 5—figure supplement 3*; Materials and methods). In the absence of stoichiometric interactions, the closer the centrosome is to the surface, the more MTs contact CFGs, which results in larger pulling forces that drives the centrosome even closer to the surface. Stoichiometric interactions prevent this destabilizing feedback. Thus, the stoichiometric interaction of MTs and CFGs allows cortical pulling forces to stably position centrosomes. Cortical pulling forces with stoichiometric interactions are stabilizing even when the number of CFGs are greater than the number of MTs.

To illustrate how a balance of pulling forces can stably position the centrosomes in the Stoichiometric Model, we simulate forces exerted on a centrosome positioned between two parallel planes with eight symmetrically arranged CFGs (four on each plane) (*Figure 5E*). We calculate the pulling force from each CFG (light purple) and its projection along the z-axis (dark purple) using the Stoichiometric Model. When the centrosome is closer to the lower plane, the pulling force from each CFG is larger on the lower plane, than on the upper, because their probability of being bound to a microtubule is greater. However, the forces from the lower CFGs are also more oblique, yielding smaller z-projection of the total pulling force from the lower plane. Thus, the net downward pulling force is smaller than the net upward pulling force, causing the centrosome to move back towards the center (*Figure 5E*, right panel). It is this change in the projection of the pulling forces, together with stoichiometry, which allows a balance of pulling forces to stably position the centrosome in the Stoichiometric Model.

A key feature of the Stoichiometric Model is that forces depend not only on the distance between a centrosome and CFGs, but depend also on the presence of a second centrosome. This occurs because when a MT from one centrosome binds an CFG, MTs from the other centrosome are prevented from binding to that CFG. This effect is evident in our simulations, where the probabilities of attachment of CFGs from one centrosome are reduced by the presence of a second centrosome (*Figure 5—figure supplement 4*) because two centrosomes compete for the CFGs between them.

A consequence of this competition is that removing one centrosome in the simulation allows the remaining centrosome to interact more strongly with CFGs at the center of the cell, which thus moves the centrosome to the cell center (*Figure 6A and B*).

To experimentally test this prediction, we removed one centrosome by laser ablating a 6×8×5.6 µm volume centered on the centrosome (*Figure 6C*). After ablation of one centrosome, the other centrosome moved to the cell center (*Figure 6D*, *Figure 6—video 1*). We repeated this experiment for 13 embryos and observed a consistent centering of the remaining centrosome (*Figure 6E*). The agreement between simulations and experiment strongly argues that the presence of one centrosome modulates the forces acting on the other centrosome, which is a key prediction of the Stoichiometric Model.

## The Stoichiometric Model explains spindle positioning, scaling, and final size

Our simulations show that pulling forces with stoichiometric interactions are sufficient to stably position centrosomes. In the simulations described above, we used a uniform density of CFGs on the cell surface. Previous studies in *C. elegans* have shown that the posterior side of the cortex has ~50% more CFGs than the anterior side (*Grill et al., 2003*; *Redemann et al., 2010*). We next modified our simulations to take into account this asymmetric distribution of CFGs. Keeping all the other parameters the same, with ~60 CFGs on the posterior side and ~40 CFGs on the anterior side, our simulations showed the centrosomes initially located ~11 µm apart near the cell center (similar to the experimentally observed metaphase spindle), move apart to a final position at a finite distance from the cell surface (*Figure 7A*). In these simulations, the posterior centrosome moves a greater distance than the anterior centrosome resulting in asymmetric positioning of the spindle center. The asymmetric motion of anterior and posterior centrosomes in our simulations is remarkably similar to the experimentally observed asymmetric motion of centrosomes in *C. elegans* embryos (*Figure 7B*). We next repeated these simulations with the same parameters but with various initial centrosome positions and various arrangements of CFGs (keeping the same total number and asymmetry) (Materials and methods). These simulations with an asymmetric distribution of CFGs accurately reproduce the dynamics of spindle elongation (*Figure 7C*), and spindle positioning (*Figure 7D*). To investigate if the Stoichiometric Model can also account for the behavior of spindle with symmetric distribution of CFGs, we studied spindles in *par-6 (RNAi)*, which greatly expands the size of the PAR-2 domain

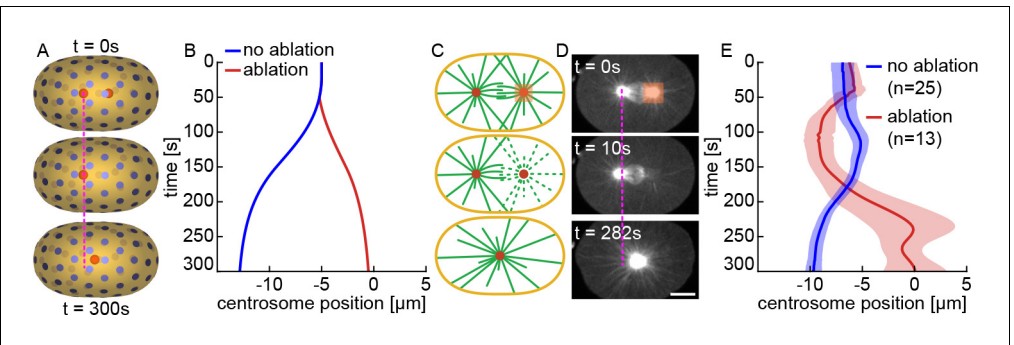

**Figure 6.** Stoichiometric Model explains centering of a single aster. (**A**) Simulation of centrosome positioning after removing one centrosome. (**B**) Centrosome position from the simulation in A (red curve), after removing one centrosome, compared to a simulation with the same parameters in presence of the other centrosome (blue curve). (**C**) Cartoon illustrating centrosome ablation and centering. (**D**) After ablation of one centrosome, the other centrosome moves to the cell center. Scale bar 10 µm. (**E**) Centrosome position as a function of time after ablating one centrosome (solid red, mean; shaded region, standard deviation, n = 13) compared to control (solid blue, mean; shaded region, standard deviation, n = 25).

The online version of this article includes the following video for figure 6:

**Figure 6—video 1.** Centrosome ablation.

https://elifesciences.org/articles/55877#fig6video1

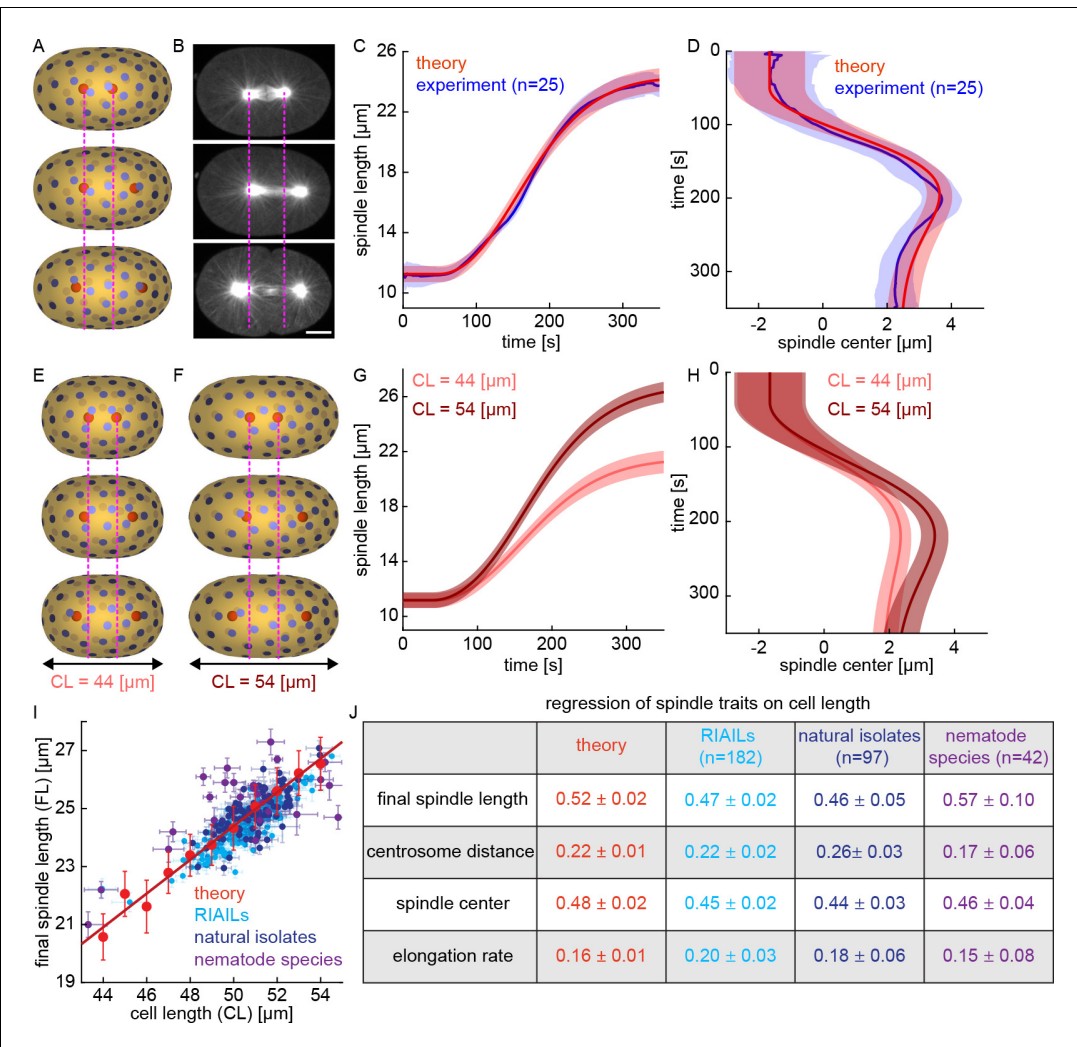

**Figure 7.** The Stoichiometric Model explains spindle elongation, positioning, and scaling with cell size. (A) Three-dimensional simulation of two centrosomes in the presence of multiple cortical CFGs with ~50% asymmetry between the right and left halves of the cell (blue disks, CFGs; red spheres, centrosomes; pink lines, position of centrosomes in the first panel). (B) Spindle elongation and centrosome movement in a *C. elegans* embryo (pink lines, position of centrosomes in the first panel). Scale bar 10 μm. (C) Red, spindle length as a function of time for multiple simulations with various initial centrosome positions and various arrangement of CFGs keeping their total number and asymmetry fixed (solid line, mean; shaded region, standard deviation); blue: spindle length as a function of time for multiple *C. elegans* embryos (solid line, mean; shaded region, standard deviation). (D) Red, spindle center as a function of time for multiple simulations with various initial centrosome positions and various arrangements of CFGs, keeping their total number and asymmetry fixed (solid line, mean; shaded region, standard deviation); blue: spindle center as a function of time for multiple *C. elegans* embryos (solid line, mean; shaded region, standard deviation). (E–F) Simulations with parameters as above but cell length is 44 μm in E and 54 μm in F (blue disks, CFGs; red spheres, centrosomes; pink lines, position of centrosomes in the first panel). (G–H) Spindle length (G) and spindle center (H) as a function of time for multiple simulations with various initial centrosome positions and various arrangement of CFGs, as above, but for cells of length 44 μm (light red) and 54 μm (dark red) (solid line, mean; shaded region, standard deviation). (I) In red, final spindle length as a function of cell length for multiple simulations with various cell length keeping the density and asymmetry of CFGs fixed (mean and standard error; red line, linear fit); final spindle length as a function of cell length across the RIAIL panel (light blue), *C. elegans* natural isolates (dark blue), and nematode species (purple) (mean and standard error). (J) Scaling of spindle traits with cell size measure by their regression on cell length for simulations (red), RIAILs (light blue), *C. elegans* natural isolates (dark blue), and nematode species (purple).

The online version of this article includes the following video and figure supplement(s) for figure 7:

*Figure 7 continued on next page*

*Figure 7 continued*

**Figure supplement 1.** The Stoichiometric Model explains spindle elongation and positioning in *par-6 (RNAi)* embryos.

**Figure 7—video 1.** Spindle elongation and positioning.

https://elifesciences.org/articles/55877#fig7video1

(*Goehring et al., 2011*). In both experiments and simulations, the spindle positioned in the middle of the cell and obtained the same final spindle length as controls (*Figure 7—figure supplement 1*).

We next investigated if the Stoichiometric Model can explain the scaling of the spindle with cell size, as well as other observed correlations between traits across the RIAILs. We simulated the dynamics of centrosomes in cells with lengths ranging from 44 $\mu m$ to 54 $\mu m$ using the same parameters as above, keeping the density of CFGs in the two halves constant (*Figure 7E and F*). In these simulations, spindles in smaller cells have a slower rate of elongation, reach a smaller final length, and position less asymmetrically (*Figure 7G and H*). This model reproduces the scaling of final spindle length with cell length observed across the RIAILs (*Figure 7I*, red, theory, regression coefficient = 0.52±0.02; light blue, RIAILs, regression coefficient = 0.47±0.02, p = 0.08). We had previously characterized the variations of spindles in ~100 *C. elegans* natural isolates and ~40 additional nematode species of known phylogeny spanning over 100 million years of evolution (*Farhadifar et al., 2015*). The scaling of final spindle length in the simulations is also in quantitative agreement with its scaling across natural isolates and across different species (*Figure 7I*, dark blue, natural isolates, regression coefficient = 0.46±0.05, p = 0.27; purple, nematode species, regression coefficient = 0.57±0.10, p = 0.62). The Stoichiometric Model also reproduces the scaling of centrosome distance from the cortex with cell length, the scaling of the final position of spindle center with cell length, and the scaling of the elongation rate with cell length across RIAILs, natural isolates, and nematode species (*Figure 7J*). Thus, the Stoichiometric Model explains the scaling of spindle traits both within and between species.

## Discussion

Previous studies have shown that spindle elongation and asymmetric positioning in *C. elegans* are driven by cortical pulling forces acting unevenly on the two centrosomes (*Grill et al., 2003*). It has been unclear if pulling forces alone can stably position centrosomes, and hence account for the final length and position of the spindle. The apparent difficulty is that pulling forces seem to be destabilizing because the closer the centrosome is to the cell periphery, the more astral MTs are expected to contact CFGs, which would imply larger pulling forces that would drive the centrosome even closer to the periphery. One proposal to circumvent this issue is that destabilizing cortical pulling forces are balanced by some other force, such as spindle elasticity or pushing from astral MTs (*Garzon-Coral et al., 2016*; *Grill et al., 2003*; *Pécréaux et al., 2016*; *Howard, 2006*; *Laan et al., 2012*; *Pavin et al., 2012*; *Grill and Hyman, 2005*; *Ma et al., 2014*; *Howard and Garzon-Coral, 2017*). Another possibility is that the magnitude of the pulling forces explicitly depends on the length of MTs, with longer MTs experiencing larger forces (*Hara and Kimura, 2009*; *Tanimoto et al., 2016*; *Tanimoto et al., 2018*; *Pierre et al., 2016*). Alternatively, intuitive arguments have been used to suggest that pulling forces alone can stably position centrosomes if there are fewer CFGs than astral MTs (*Grill and Hyman, 2005*). Our laser ablation experiments and genetic perturbations show that spindle elasticity and pushing forces do not significantly contribute to the final position of centrosomes in *C. elegans* embryos.

We developed the Stoichiometric Model of cortical pulling forces, which is based on known biochemical properties of MTs and molecular motors. In this model, each CFG can only bind one MT at a time (i.e. the interactions are stoichiometric), and every MT that is attached to a CFG, experiences the same magnitude of pulling force, irrespective of the MT's length. The probability that MTs contact a CFG increases with decreasing distance between the centrosome and the CFG, and thus the average force that the CFG exerts on the centrosome depends on their distance. In this model, the stable positioning of centrosomes results from the stoichiometric interaction between MTs and CFGs, which prevents the destabilizing feedback present in previous models of cortical pulling forces. When the stoichiometric interactions are present, cortical pulling forces can be stabilizing

even when there are more CFGs than MTs. Thus, cortical pulling forces are sufficient to stably position centrosomes even in the absence of explicit length dependent forces or a limiting number of CFGs.

In *C. elegans*, the spindle is asymmetrically positioned during anaphase, which causes asymmetric formation of the furrow and cell division. The Stoichiometric Model quantitatively explains the asymmetry in positioning of the spindle center by an uneven distribution of CFGs between the anterior and posterior halves of the embryo. In our simulations, we considered two domains with ~50% enrichment of CFGs on the posterior half compared to the anterior. Although a 'two-domain' model has been widely used to describe asymmetric positioning of the spindle in *C. elegans*, a more detailed 'three-domain' model has also been proposed (*Krueger et al., 2010*). While the 'two-domain' CFG model is sufficient to quantitatively explain the experimental results in the present manuscript, an interesting future direction would be to expand this model to incorporate more complex distribution of CFGs and attempt to explain the consequence of other perturbations, such as LET-99 knockdown.

Recombinant inbred lines provide a powerful means to generate quantitative variations in diverse biological traits. Investigating the correlations and partial correlations between these traits, and mapping their genetic bases, provides a systematic approach to disentangle their relations (*Rockman, 2008*). Such approaches have been used to study gene expression (*Rockman et al., 2010*; *Keurentjes et al., 2007*; *Chick et al., 2016*) and a wide variety of complex physiological processes (*Pitchers et al., 2019*; *Greene et al., 2016*; *Andersen et al., 2014*; *Beamer et al., 2001*; *Linnen et al., 2013*). We used a similar methodology to test general classes of models of spindles in *C. elegans* embryos. After identifying a class of models consistent with the data, and relevant genetic factors, we employed biophysical experiments to investigate the forces acting on the spindle. These experimental results led us to develop the Stoichiometric Model, a mechanistic mathematical model of the coordination of spindle elongation, positioning, and cell size. Our approach of combining quantitative genetics and biophysics can be adopted to study the regulation and coordination of other complex cell biological processes.

In a previous study (*Farhadifar et al., 2015*), we characterized the first mitotic division in ~100 *C. elegans* natural isolates and ~40 additional nematode species and discovered extensive variations in spindles, both within and between species. Variations in all aspects of spindles we investigated were correlated with cell length. We found evidence that cell length is subject to stabilizing selection, which, due to the correlation of spindle traits with cell length, is sufficient to explain the variations in spindles within and between species. Thus, the evolution of the spindle in nematodes is primarily driven by correlations between the spindle and cell length, but it was unclear what cell biological processes produced these correlations. As part of the present study, we investigated the correlations between spindles and cell length across the recombinant inbred lines and found that they are the same as the correlations across natural isolates and nematode species. The Stoichiometric Model we developed quantitatively reproduces these correlations. This provides a mechanistic explanation for the evolution of the spindle in nematodes.

## Materials and methods

### Experimental procedures

#### Maintenance and time-lapse microscopy of the recombinant inbred advanced intercross lines (RIAILs)

We cultured the RIAILs at 24°C on nematode growth media (NGM) plates and fed with *Escherichia coli* OP50 as described previously (*Brenner, 1974*). After thawing (~20 lines simultaneously), we propagated the RIAILs for approximately 1 week, followed by dividing each RIAIL into five replicate plates. We then shuffled these plates in the incubator and picked one plate at random for microscopy. We dissected adult worms in M9 buffer, mounted embryos on a 4% agar pad between a slide and a coverslip and used an eyelash to position multiple embryos into close proximity. We performed differential interference contrast (DIC) microscopy on a Nikon Eclipse TE2000-E microscope equipped with a 40x Plan Apochromat NA 1.25 objective and an oil-immersed condenser NA 1.4. Every second, we acquired 13 z-planes separated by 1 μm using a Hamamatsu ORCA-R2 camera and a piezo-driven nanopositioning stage Physik Instrumente E-709.

## Image processing and quantification of spindle traits across the RIAILs

We used custom-designed image-processing software as described previously (*Farhadifar and Needleman, 2014*) to segment and track spindle poles in DIC images of *C. elegans* embryos. For each embryo, we measured spindle length - the distance between the two spindle poles - as a function of time and fitted a sigmoid $l = IL + (FL - IL)/(1 + \exp(-(t - t_0)/\tau_s))$ to the data, where $IL$ is the initial spindle length, $FL$ is final spindle length, $\tau_s$ is the characteristic time of spindle elongation, and $t_0$ is the time at which the spindle is elongating at its maximum rate. We defined elongation rate, $ER$, to be the rate of spindle elongation at $t = t_0$, which is given by $(FL - IL)/4\tau_s$. We fit the measured distance of the posterior centrosome from the posterior edge of the cell using a sigmoid $d = CD + d_1/(1 + \exp((t - t_1)/\tau_C))$, where $CD$ is the final centrosome distance. We measured cell area as the area enclosed by the embryo at the end of cell division. We defined cell length, CL, as the distance between the anterior and posterior ends of the cell at the end of cell division.

## QTL mapping of RIAILs

We performed QTL mapping using R/qtl (*Broman and Sen, 2009*). The RIAIL panel consists of two sets of lines derived from inbreeding hermaphrodites in the tenth generation of the cross (*Rockman and Kruglyak, 2009*). Linkage scans were performed separately for the two subsets, and LOD scores were then summed, and p-values were estimated from 500 permutations performed independently for the two subgroups. For mapping final spindle length conditioned on cell length and other QTLs, we used linear regression.

## Fluorescence imaging and RNA interference

Strain SA250 (tjIs54 [pie-1p::GFP::tbb-2 + pie-1p::2xmCherry::tbg-1 + unc-119(+)]; tjIs57 [pie-1p::mCherry::his-48 + unc-119(+)]) was used for experiments with fluorescence imaging, laser ablation, and RNA interference. We cultured SA250 at 24°C on nematode growth media (NGM) plates and fed with *Escherichia coli* OP50 as described previously (*Brenner, 1974*). For imaging the spindle, we dissected adult worms in M9 buffer and mounted embryos on a 4% agar pad between a slide and a coverslip. RNA interference (RNAi) was carried out following the RNAi feeding protocol (*Kamath et al., 2000*). For *gpr-1/2 RNAi*, we fed L2 worms on the RNAi bacterial lawn at 24°C for 48 hr before imaging. For *spd-1 RNAi*, we fed young L4 worms at 24°C for 24 hr on the RNAi bacterial lawn before imaging. For *par-2 RNAi*, we fed young L4 worms at 24°C for 36 hr on the RNAi bacterial lawn before imaging.

## Spinning disk confocal fluorescence imaging

For live fluorescent imaging of the spindle, we used a spinning disk confocal microscope (Nikon TE2000, Yokugawa CSU-X1), equipped with 488 nm and 561 nm diode lasers, an EMCCD camera (Hamamatsu), and a 60X water-immersion objective (CFI Plan Apo VC 60X WI, NA 1.2, Nikon). We used a home-developed LabVIEW program (LabVIEW, National Instruments) to control the parameters of the imaging.

## Laser ablation of spindle and centrosomes

For laser ablation, we used a custom-build system with a femtosecond near-infrared Ti:sapphire pulsed laser (Mai-Tai, Spectra-Physics, Mountain View, CA) and a pulse picker (Eclipse Pulse Picker, KMLabs) to generate a 16 kHz femtosecond pulse train with ~6-nJ pulse energy. We used the imaging objective to focus the laser to a diffraction limited spot, which was scanned over the sample in three dimensions at speeds between 150 and 200 µm/s to ablate-defined geometries using a piezostage (P-545 PInano XYZ, Physik Instrumente) and home-developed LabVIEW software (LabVIEW, National Instruments).

## Processing fluorescent images

For all experiments with fluorescent microscopy, we manually tracked the centrosomes with ImageJ. We used custom MATLAB code to align spindle length curves from different embryos and calculated the motion of centrosomes.

## Measuring correlation and partial correlation between traits

To measure the correlation coefficient between two quantitative traits $T_1$ and $T_2$, we fit a linear model to the data, $T_2 = m \times T_1 + b$, and extract the regression coefficient $m$ and the 95% prediction intervals of the fit. To measure the partial correlation between quantitative traits $T_1$ and $T_2$ conditioned on trait $T_3$, we first measure the residual of linear fits of $T_1$ on $T_3$ and $T_2$ on $T_3$:

$$R_1 = T_1 - m_1 \times T_3 - b_1$$

$$R_2 = T_2 - m_2 \times T_3 - b_2$$

where $m_1$ and $b_1$ are the slope and intercept of linear regression of $T_1$ on $T_3$, and $m_2$ and $b_2$ are the slope and intercept of linear regression of $T_2$ on $T_3$. We then fit a linear model to the residuals $R_2 = s \times R_1 + c$ and extract the regression coefficient $s$ and the 95% prediction intervals of the fit.

## Theoretical procedures

In this section, we construct a model of cortical pulling forces based on the known biochemical properties of microtubules and molecular motors. This derivation is done in steps: first, we consider the dynamics of microtubules nucleating and growing from a centrosome; second, we analyze microtubules growing toward a single force-generator; third, we calculate the force that a force-generator exerts on a centrosome, with and without stoichiometric interactions, and how this is modified by the presence of a second centrosome; finally, we derive the equation of motion for centrosomes in the presence of multiple force-generators.

### Nucleation and growth of microtubules from a centrosome

We consider a centrosome with microtubules nucleating with equal probability in all directions with rate $\gamma$, which then grow with velocity $V_g$ and undergo catastrophe with rate $\lambda$. The length distribution of microtubules, $\psi(l,t)$, satisfies the Fokker-Planck equation

$$\partial_t \psi(l,t) + V_g \partial_l \psi(l,t) = -\lambda \psi(l,t) \tag{1}$$

To impose a constant nucleation rate at the centrosome, we set the boundary condition $\psi(0,t) = \frac{\gamma}{V_g}$. Solving this in steady state, in the absence of boundaries, gives the length distribution of microtubules as $\psi(l) = \frac{\gamma}{V_g} e^{-l\lambda/V_g}$.

The total number of microtubules, $N_{MT}(t)$, is given by

$$N_{MT}(t) = \int_0^\infty \psi(l,t)dl \tag{2}$$

The time derivative of microtubule number in the absence of boundaries is

$$\begin{aligned}
\frac{dN_{MT}(t)}{dt} &= \int_0^\infty \partial_t \psi(l,t)dl = -V_g \int_0^\infty \partial_l \psi(l,t)dl - \lambda \int_0^\infty \psi(l,t)dl \\
&= -V_g(\psi(\infty,t) - \psi(0,t)) - \lambda N_{MT}(t) = -V_g(0 - \gamma/V_g) - \lambda N_{MT}(t) \\
&= \gamma - \lambda N_{MT}(t)
\end{aligned} \tag{3}$$

Solving this equation in steady state gives the total number of microtubules as $N_{MT} = \gamma/\lambda$.

### Rate at which growing microtubules impinge upon a force-generator

We next consider a centrosome located at the origin and a disk-shape force-generator of radius $r$, a distance $d$ away, with an outward unit normal $\hat{n}$ (*Figure 5—figure supplement 5*). Here, we calculate the rate that growing microtubules impinge upon the force-generator. Only microtubules located in a cone defined by the position of the centrosome and projected area of the force-generator can grow to contact the force-generator. The number of microtubules located in this cone is:

$$N(t) = \hat{\xi} \cdot \hat{n} \int_0^{\upsilon_m} \frac{\sin\upsilon}{2} d\upsilon \int_0^d \psi(l,t)dl = \chi(d) \int_0^d \psi(l,t)dl \tag{4}$$

where $\hat{\xi}$ is the unit vector pointing from the centrosome to the force-generator, $\vartheta_m = \tan^{-1}(r/d)$ is

the solid angle of the cone, and $\chi(d) = \frac{\hat{\xi}\cdot\hat{n}}{2}\left(1 - \frac{1}{\sqrt{1+(r/d)^2}}\right)$ is the fraction of microtubules nucleated from the centrosome that fall inside the cone.

For a fixed distance, $d$, the time derivative of the number microtubules inside the cone is

$$\frac{dN(t)}{dt} = \chi(d)\int_0^d \partial_t \psi(l,t)dl = \chi(d)\big(\gamma - \lambda N(t) - V_g\psi(d,t)\big) \tag{5}$$

The first and second terms on the right-hand side are the rates that microtubules are generated by nucleation and that disappear by catastrophe. The third term is the rate microtubules leave the cone by contacting the force-generator. Solving this gives the rate that microtubules impinge upon the force-generator, $\Omega(d)$, in steady state, as:

$$\Omega(d) = V_g\chi(d)\psi(d) = \gamma\chi(d)e^{-d\lambda/V_g} \tag{6}$$

## Force-generators with stoichiometric interactions

We next calculate the pulling force on a centrosome from a force-generator a distance $d$ away. We considered two scenarios: first, when the interaction of microtubules and the force-generator is stoichiometric, that is only one microtubule can bind to a force-generator at a time; second, when the interaction of microtubules and the force-generator is non-stoichiometric, that is all the microtubule that reach the force-generator bind to it. In both scenarios, the microtubules that bind to the force-generator are subject to pulling force as long as they are bound.

We first calculate the force on centrosome when the interaction of microtubules and the force-generator is stoichiometric. At any given instant, the force on the centrosome is $f_0\hat{\xi}$ if a microtubule is bound to the force-generator, and zero otherwise. Here, we consider the scenario in which the centrosome moves slowly compared to the polymerization dynamics of microtubules. In that case, at time-scales longer than the life-time of an individual microtubule, the pulling force, $\vec{F}$, on the centrosome is proportional to the probability of attachment of a microtubule to the force-generator, $P$, and is given by:

$$\vec{F} = f_0 P\hat{\xi} \tag{7}$$

Here, $P$ obeys the dynamics:

$$\frac{dP(d,t)}{dt} = \Omega(d)(1 - P(d,t)) - \kappa P(d,t) \tag{8}$$

where $\kappa$ is the rate of microtubule detachment from the force-generator. If the motion of centrosome is slow compared to the binding and unbinding dynamics, then the attachment probability can be approximated as quasi steady-state with $P(d) = \Omega(d)/(\Omega(d)+\kappa)$. Thus, the force acting on the centrosome is

$$\vec{F} = f_0\frac{\Omega(d)}{\Omega(d)+k}\hat{\xi} \tag{9}$$

We next considered how stoichiometric pulling forces exerted on a centrosome are modified by the presence of a second centrosome. This occurs because microtubules from the second centrosome can transiently bind to the force-generator and thereby temporarily block binding by microtubules from the first centrosome. This leads to a competition between centrosomes for occupation of force-generator. Therefore, the presence of the second centrosome will result in a decrease in $P$ from the first centrosome, leading to a reduction of force on the first centrosome. If centrosome 1 is located at distance $d_1$ from the force-generator, the force acting on it is $\vec{F}_1 = f_0 P_1\hat{\xi}_1$, with:

$$\frac{dP_1(d_1,t)}{dt} = \Omega(d_1)(1 - P_1(d_1,t) - P_2(d_2,t)) - \kappa P_1(d_1,t)$$

$$\frac{dP_2(d_2,t)}{dt} = \Omega(d_2)(1 - P_1(d_1,t) - P_2(d_2,t)) - \kappa P_2(d_2,t) \tag{10}$$

where $d_2$ is the distance of the second centrosome to the force-generator, and $P_2(d_2, t)$ is the probability of attachment of microtubules from the second centrosome. If the motion of both centrosomes are slow compared to the binding and unbinding dynamics, then the attachment probability can be approximated as in quasi steady-state with $P_i(d_i) = \Omega(d_i)/(\Omega(d_1) + \Omega(d_2) + \kappa)$ with $i = 1, 2$. Therefore, in a presence of the second centrosome, the force on the first centrosome is:

$$\vec{F}_1 = f_0 \frac{\Omega(d_1)}{\Omega(d_1) + \Omega(d_2) + k} \hat{\xi}_1 \tag{11}$$

Comparing this to the formula above for the force with only one centrosome, shows a reduction in the force due to the extra factor of $\Omega(d_2)$ in the denominator, which results from competition between centrosomes for occupation of the force-generator.

## Force-generators with non-stoichiometric interactions

We next derive a model of non-stoichiometric interactions in which any microtubule that contacts a force-generator is subject to pulling forces, even if that force-generator is already pulling on other microtubules. In this model, the pulling force that a force-generator exerts on a centrosome is proportional to the average number of microtubules from that centrosome that contact the force-generator. If the motion of centrosome is slow compared to the binding and unbinding dynamics, then the average number of bound microtubules is $\bar{n} = \Omega(d)/\kappa$ and the pulling force is

$$\vec{F} = f_0 \bar{n} \hat{\xi} = f_0 \frac{\Omega(d)}{k} \hat{\xi} \tag{12}$$

For non-stoichiometric interactions, there is no competition between centrosomes and the presence of a second centrosome does not alter the force that the force-generator exerts on the first centrosome.

In the experimentally relevant regime, the nucleation rate $\gamma$ is much larger than the detachment rate $\kappa$ ($\gamma \sim 250 \ s^{-1}$ and $\kappa \sim 0.01 \ s^{-1}$). In this limit, as the centrosome gets close to the force-generator and $d \to 0$, linear expansion of the derived forces shows that the pulling force with non-stoichiometric interaction increases as $\gamma/\kappa$, while the pulling force with stoichiometric interactions increases as $\kappa/\gamma$. Thus, the increase in force as $d \to 0$ occurs $(\kappa/\gamma)^2 \sim 10^{-8}$ slower for stoichiometric interactions. This occurs because in the absence of stoichiometric interactions, the closer the centrosome is to the force-generator, the more microtubules contact it, which results in even larger pulling forces. Stoichiometric interactions prevent this destabilizing feedback.

## Centrosome dynamics in presence of multiple force-generators

In this section, we construct the equations of motion for two centrosomes in the presence of multiple force-generators with stoichiometric interactions in the limit that centrosome motion is slow compared to microtubules polymerization and binding dynamics. Considering $M$ non-overlapping force-generators distributed on the cell periphery with their centers positioned at $\vec{X}_{m,j}$ ($j = 1, 2, \ldots, M$). The total forces on the two centrosomes located at $\vec{X}_i$ ($i = 1, 2$) are:

$$\vec{F}_i = \sum_{j=1}^{M} \vec{F}_i^j = \sum_{j=1}^{M} f_0 P_i^j \hat{\xi}_i^j \tag{13}$$

where $P_i^j$ is the probability of attachment of a microtubule from centrosome $i$ to force-generator $j$, and $\hat{\xi}_i^j$ is the unit vector from centrosome $i$ to force-generator $j$.

In addition to pulling forces from the force-generators, we considered two other forces: drag on the centrosomes, $\eta$, with drag coefficient $\vec{f}_{drag,i} = \eta \dot{\vec{X}}_i$; the force from the central spindle, which we model as a viscous element directed along the spindle axis $\hat{S} = (\vec{X}_1 - \vec{X}_2)/|\vec{X}_1 - \vec{X}_2|$, with viscous friction coefficient $\upsilon$. Thus, force-balance gives

$$\begin{aligned} \eta \dot{\vec{X}}_1 + \upsilon(\dot{\vec{X}}_1 - \dot{\vec{X}}_2) \cdot \hat{S}\hat{S} = \vec{F}_1 \\ \eta \dot{\vec{X}}_2 - \upsilon(\dot{\vec{X}}_1 - \dot{\vec{X}}_2) \cdot \hat{S}\hat{S} = \vec{F}_2 \end{aligned} \tag{14}$$

These equations, along with the pulling force model described above, gives the equations of motion for centrosomes once the cell shape and positions of force-generators are set.

## Cell shape and force-generator distribution

We modeled the geometry of the *C. elegans* embryo as a superellipsoid

$$\left(\frac{x}{R_x}\right)^{P_x} + \left(\frac{y}{R_y}\right)^{P_y} + \left(\frac{z}{R_z}\right)^{P_z} = 1 \tag{15}$$

where $R_x$, $R_y$, and $R_z$ are the radii in the $x$, $y$, and $z$ directions, and $P_x$, $P_y$, and $P_z$ define the axial curvatures. We used $R_y = R_z = 15\,\mu m$, with $R_x = 25\,\mu m$ for all simulations except those in *Figure 7E and F*, where we changed $R_x$ accordingly. For all simulations, we used $P_x = 3$, $P_y = 2$, and $P_z = 2$.

We used DistMesh (*Persson and Strang, 2004*) to generate a uniform distribution of force-generators on each half of the cell surface, with a given asymmetry. To determine the robustness of the simulation to precise motor positions, we introduced randomness in the positioning of force-generators by slightly displacing them from their positions set by DistMesh. We ran simulations with 64 different motor configurations and different initial centrosome positions, and quantified the average and standard deviation of centrosome dynamics.

Following *Pecreaux et al., 2006*, we modeled motor properties as changing with time during anaphase. We varied $f_0$ linearly with time. Using time-independent motor properties produces the same final spindle length, final spindle position, elongation rate, and scaling. However, time-independent motor properties produce a discontinuity in the rate of spindle elongation and positioning at the transition between metaphase and anaphase.

## Simulation procedure

We initialize the simulation by specifying the number, location, and orientation of force-generators as described in the section 'Cell shape and force-generator distribution', and the starting position of the centrosomes. At each time step of the simulation, we first calculate the force exerted by each force-generator on the centrosomes using *Equation 6* for the impingement rate of microtubules on that force-generator and *Equation 11* to account for the stoichiometric interactions through the probability of attachment. We then use *Equation 13* to calculate the net pulling force on each centrosome and use *Equation 14* to update the position of centrosomes accounting for drag on the centrosomes and central spindle viscosity.

## Testing the 'limited force-generator' hypothesis

To test if the stable positioning of centrosomes in the Stoichiometric Model is due to the limited number of CFGs relative to the number of microtubules, we simulated spindle elongation for various number of CFGs (up to 100,000 CFGs; ~10 times larger than the number of microtubules). Using DistMesh, we first generated configurations of evenly distributed CFGs with $N = 100$, 1000, 10,000 and 100,000. To prevent overlaps between CFGs at large values of $N$, we will need to substantially reduce $r$ (in the previous simulations, we use $r = 1.5\mu m$), while keeping the elongation dynamics fixed. *Figure 5—figure supplement 2A and B* show, that for fixed $N = 100$, the elongation dynamics can be conserved by decreasing the detachment rate $\kappa$ as $r$ is decreased (all other parameters are held fixed; varying $r$ alone affects both spindle elongation and final size, see *Figure 5—figure supplement 1D*). In particular, we found that the Stoichiometric Model generates similar spindle elongation and final size for $\kappa = 4.4E - 4[s^{-1}]$ with $r = 0.1\mu m$, compared to the previous simulations using $\kappa = 0.1[s^{-1}]$ with $r = 1.5\mu m$ (*Figure 5—figure supplement 2A and B*). We then simulated spindle elongation for $r = 0.1\mu m$ and $\kappa = 4.4E - 4[s^{-1}]$, changing simultaneously the number of CFGs ($N = $ 100, 1000, 10,000 and 100,000) and the force per CFG, $f_0$, so that $Nf_0$ is fixed (varying $N$ alone in the model affects spindle elongation dynamics; see *Figure 5—figure supplement 1C*). We found similar spindle elongation dynamics and final size as $N$ was increased, and as seen in particular for $N = 100,000$ and $f_0 = 0.01pN$, where we have 1000 times more CFGs than in the original simulations with $N = 100$ (*Figure 5—figure supplement 2C and D*). Thus, a limited number of CFGs does not explain the stable positioning of centrosomes and spindle final length in the Stoichiometric Model.

## Parameters used in simulations

We use the following parameters for the simulations, unless noted otherwise:

| Simulation parameter | Value | Reference |
|---|---|---|
| Microtubule growth rate ($V_g$) | 0.5 [$\mu m/s$] | *Srayko et al., 2005* |
| Microtubule catastrophe rate ($\lambda$) | 0.025 [$s^{-1}$] | *Kozlowski et al., 2007* |
| Microtubule nucleation rate ($\gamma$) | 250 [$s^{-1}$] | *Redemann et al., 2017* |
| Microtubule-force-generator detachment rate ($\kappa$) | 0.1 [$s^{-1}$] | *Redemann et al., 2010* |
| Force-generator capture radius ($r$) | 1.5 $\mu m$ | *Kozlowski et al., 2007*; *Gusnowski and Srayko, 2011* |
| Force-generator pulling force ($f_0$) | 10 $pN$ | *Kozlowski et al., 2007* |
| Centrosome drag ($\eta$) | 150 $pNs/\mu m$ | *Garzon-Coral et al., 2016* |
| Spindle viscous friction coefficient ($\nu$) | 100 $pNs/\mu m$ | * |

* Estimated in this study.

## Acknowledgements

We thank JA Calarco, E Nazockdast, and D Riccardi for useful discussions and assistance. We acknowledge the Caenorhabditis Genetics Center (CGC) for providing us with some strains used in this study. The CGC is funded by the NIH Office of Research Infrastructure Programs (P40 OD010440). The computations in this paper were run, in part, on the Odyssey cluster supported by the FAS Division of Science, Research Computing Group at Harvard University, and were run, in part, on facilities supported by the Scientific Computing Core at the Flatiron Institute. Support was provided by Human Frontier Science Program grant RGP 0034/2010 to TM-R, and DJN, National Science Foundation Grants DBI-0959721 and DBI-1919834 to DJN, National Institutes of Health Grant 1R01GM104976-01, and National Science Foundation under awards DMR-1420073 (NYU MRSEC), DMS-1620331, and DMR-2004469 to MJS, and NIH Grant 1R01GM121828 to MVR. TM-R received funding from the German Research Foundation (DFG grants MU 1423/8–1 and 8–2).

## Additional information

### Funding

| Funder | Grant reference number | Author |
|---|---|---|
| Human Frontier Science Program | RGP 0034/2010 | Thomas Müller-Reichert Daniel J Needleman |
| National Science Foundation | DBI-0959721 | Daniel J Needleman |
| National Institutes of Health | 1R01GM104976-01 | Michael J Shelley |
| National Institutes of Health | 1R01GM121828 | Matthew Rockman |
| Deutsche Forschungsgemeinschaft | MU 1423/8-1 | Thomas Müller-Reichert |
| Deutsche Forschungsgemeinschaft | MU 1423/8-2 | Thomas Müller-Reichert |
| National Science Foundation | DBI-1919834 | Daniel J Needleman |
| National Science Foundation | DMR-1420073 (NYU MRSEC) | Michael J Shelley |
| National Science Foundation | DMS-1620331 | Michael J Shelley |
| National Science Foundation | DMR-2004469 | Michael J Shelley |

The funders had no role in study design, data collection and interpretation, or the decision to submit the work for publication.

## Author contributions
Reza Farhadifar, Conceptualization, Resources, Software, Formal analysis, Validation, Investigation, Visualization, Methodology, Writing - original draft, Writing - review and editing; Che-Hang Yu, Hai-Yin Wu, Resources, Methodology; Gunar Fabig, Resources, Formal analysis, Investigation, Methodology, Writing - review and editing; David B Stein, Software; Matthew Rockman, Resources, Software, Formal analysis, Investigation, Writing - review and editing; Thomas Müller-Reichert, Conceptualization, Supervision, Funding acquisition, Writing - review and editing; Michael J Shelley, Conceptualization, Formal analysis, Supervision, Investigation, Writing - original draft, Writing - review and editing; Daniel J Needleman, Conceptualization, Resources, Formal analysis, Supervision, Funding acquisition, Investigation, Writing - original draft, Writing - review and editing

## Author ORCIDs
Reza Farhadifar https://orcid.org/0000-0003-2792-5380
Che-Hang Yu https://orcid.org/0000-0002-0353-9752
Gunar Fabig http://orcid.org/0000-0003-3017-0978
Matthew Rockman http://orcid.org/0000-0001-6492-8906
Thomas Müller-Reichert http://orcid.org/0000-0003-0203-1436

## Decision letter and Author response
Decision letter https://doi.org/10.7554/eLife.55877.sa1
Author response https://doi.org/10.7554/eLife.55877.sa2

# Additional files

## Supplementary files
- Source code 1. Stoichiometric model.
- Transparent reporting form

## Data availability
With our manuscript, we submitted a sample movie for high-throughput microscopy of embryos from the *C. elegans* recombinant inbred panel (Figure 1—video 1) and a sample movie for segmentation and tracking of the first mitotic spindle in these embryos (Figure 1—video 2). The raw movies for the whole panel are ~12,000,000 images (~17 TB), which is a large dataset for public servers. However, we can transfer the data upon request.

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
