## [Decision Letter]

**Acceptance summary:**

This work presents a tour de force, imaging 182 lines of *C. elegans* embryos during the first division to infere their scaling and positioning, and high-precision laser ablation to examine the forces acting on the spindle. The authors have generated a quantitative Stoichiometric Model, where force generators located on the cell cortex pull on astral microtubules with each force generator pulling on one microtubule at most. This model explains the dynamics and variability of spindle scaling across different nematodes, representing variations over millions of years of evolution.

**Decision letter after peer review:**

Thank you for submitting your article "Stoichiometric interactions explains spindle dynamics and scaling across 100 million years of nematode evolution" for consideration by *eLife*. Your article has been reviewed by three peer reviewers, including Julie P I Welburn as the Reviewing Editor and Reviewer #1, and the evaluation has been overseen by Anna Akhmanova as the Senior Editor.

The reviewers have discussed the reviews with one another and the Reviewing Editor has drafted this decision to help you prepare a revised submission.

Farhadifar et al. propose a model for spindle scaling across *C. elegans* species, where one cortical force generator binds to one microtubule to center the spindle. They first perform a high throughput analysis of the first division of embryo, quantifying cell length, spindle length and other features and examining the variations across different lines. They find that spindle length is dependent on cell length, but not with other cell parameters like the initial spindle size or cell volume. They test different models for regulation of spindle length and rule them out. They then identify genes associated with traits that control spindle length as GPR1/2 and PAR proteins, which is not novel. These complexes are known cortical force generator pulling on the centrosomes through astral microtubules. They then perform laser ablation experiments on spindles that have reached their final length to examine the balance and origin of forces on the centrosomes and rule out possible ways the spindle length is set. These data feed into a mathematical model termed thereafter "stoichiometric model" where one cortical force generator can bind only one microtubule. They propose the stoichiometric interaction is sufficient and a requirement to stably center the spindle. Finally, the authors extend their model to other nematodes by comparing theoretical final spindle length predicted by the stoichiometric model to previously published data from experimentally measured spindle length in *C. elegans* natural isolates and other nematode species.

The reviewers were all supportive of asking major revisions, described below. The feeling was that the authors need to improve the clarity and accuracy of this manuscript, especially the section related to the model, while shortening it during revision. Please pay special attention to point 9. Appropriate referencing and acknowledgements to others' work should be included.

1) Some of the authors' conclusions disagree with some previously published findings and models (e.g. Astral microtubules generate pushing forces on spindle poles in Garzon-Corral et al., 2016), however the authors do not provide any potential explanation/hypothesis for this discrepancy. Could it be that the laser ablation experiments, which is the primary experimental base on which the authors draw their conclusions in this manuscript, have some intrinsic caveats (incomplete ablation of microtubules or side-effects of the ablation or rapid repolymerization of spindle microtubules or else) that would preclude drawing reliable conclusions? At minimum, it seems that some important controls are missing to draw strong conclusions (e.g. genetic evidence to back up their laser-ablation experiments and immunofluorescent staining of microtubules following ablation to demonstrate efficiency, etc). The only genetic control provided (i.e. spd-1(RNAi) compared to spindle microtubule ablation) is not really convincing as the two centrosomes clearly overshoot compared to controls following central spindle breakage, and the spindle over-elongates before going back to a control length (Figure 4I: the bump on the red curve). Would this be predicted by the stoichiometric model?

2) All the ablation experiments that are presented in this manuscript have been performed on the posterior half of the spindle. Since the stoichiometric model hypothesize that the two centrosomes are positioned by an identical mechanism (i.e. by a balance of cortical pulling forces), another important set of control experiments would be to ablate the different microtubule populations on the anterior side of the spindle.

3) The stoichiometric model posits that centrosomes are positioned by a balance of pulling forces acting in different directions. This is not consistent with genetic evidence on the role of LET-99, which, in the posterior half of the embryo, restricts cortical pulling forces to the posterior-most region of the cortex. How do the authors reconcile this discrepancy between the design of their mathematical model and this genetic data?

4) Another important parameter in the stoichiometric model is microtubule dynamics. In particular, the catastrophe frequency of astral microtubules is likely to have a major effect on the outcome of the simulation. The parameters (growth rate, catastrophe frequency and nucleation rate) used in the simulation are listed in a table at the end of the manuscript but it is not clear how these values have been set.

5) A model of spindle positioning in *C. elegans* that relies solely on cortical pulling forces, and which the authors omitted to cite in their current manuscript, has already been published (Bouvrais et al., 2018). This previous model has the advantage that it fully accounts for the effect of the LET-99 loss-of-function observed in vivo on spindle positioning and elongation. Does the model in the current manuscript represents a significant advance in our understanding of spindle elongation and positioning?

6) The fact that gpr-1/2 and par-2 are important regulators of spindle elongation during anaphase, and thus of final spindle length has been demonstrated many years ago. Therefore I'm not convinced that the QTL analysis performed in this study really highlights a novel feature of the mechanism that controls spindle elongation in the one-cell *C. elegans* embryo.

7) They carry out some RNAi experiments of the gpr1 and par mutants to examine spindle length. Under these conditions, one loses the cortical interaction to the microtubule. What does their model predict in the absence of cortical forces in terms of spindle length?

8) The authors do not give details about their stochastic simulation. I am still not sure what they really did. Did they simulate microtubules growing out of centrosomes and then calculate the forces to obtain the centrosome dynamics? Or did they use the probabilities for microtubule attachment to a force generator and based their simulation on these probabilities?

9) The authors did not do a good job in explaining the mechanism by which the stoichiometric model leads to centrosome centering. It is still true that the probability of a force generator binding a microtubule increases with the centrosome coming closer to the force generator. The authors state that "In this model, the stable positioning of centrosomes results from the stoichiometric interaction between MTs and CFGs, which prevents the destabilizing feedback present in previous models of cortical pulling forces.". This did not help to understand, why the model works. It seems that the central ingredient is that the forces are along the direction of the microtubules and that these forces vary in magnitude. This is because the force generated by the force generator is perpendicular to the cell boundary and it is this force that is the same in magnitude for all force generators; the component projected into the direction of the microtubule is not. As the centrosome moves to or away from the boundary, the angle of the microtubules with the boundary and thus the force they experience changes. If I understood correctly, this is why the centrosome position is stabilized: the closer you get to the boundary the smaller the component in the direction of the microtubule typically becomes (the simplest situation to see this mechanism at work is a centrosome between two parallel plates, where there are two force generators on each plate). The condition of having at most one microtubule bound to a force generator prevents the binding of more and more microtubules to the force generator nearest to the centrosome, which would lead to an instability of the central centrosome position. – Independently of whether the mechanism I invoke here is correct or not, I would urge the authors to explain the mechanism of stabilizing the central centrosome position and not just to state that their mechanism does so.

[Editors' note: further revisions were suggested prior to acceptance, as described below.]

Thank you for submitting your article "Stoichiometric interactions explains spindle dynamics and scaling across 100 million years of nematode evolution" for consideration by *eLife*. Your article has been reviewed by two peer reviewers, and the evaluation has been overseen by a Reviewing Editor and Anna Akhmanova as the Senior Editor. The reviewers have opted to remain anonymous.

The reviewers have discussed the reviews with one another and the Reviewing Editor has drafted this decision to help you prepare a revised submission.

The reviewers agreed that the manuscript was improved, yet a number of points raised by the reviewers have not been addressed in the text. Please go through each point (in previous decision letter) and rather than only respond to the reviewers, use their comments to guide the editing of your manuscript to improve clarity of the text and mention the changes to the text in the rebuttal letter, shortening the text where appropriate to streamline. In the title, "interactions" is plural ut the verb is singular, this needs correcting. Please also discuss your results in the light of other people's work-see below.

Additional Major comments

1) “We believe there is no discrepancy between our results and previous published findings. The work of Garzon-Corral et al., 2016, is focused on forces in metaphase. Our study addresses forces on centrosomes at the end of anaphase. It is possible that different forces dominate at different times, which will be interesting and important to investigate in future studies.”

The work of Garzon-Coral et al., 2016, focuses on forces in metaphase AND anaphase (see Figure 3 in their paper). One of the main conclusions from Garzon-Coral et al., 2016, is: “During anaphase, forces on the order of 100 pN were required to displace the spindle 1 mm. These forces are similar in magnitude to the forces measured during chromosome segregation by Nicklas in grasshopper cells (6). An increase in the centering force may help to stabilize spindle position against high centrifugal forces that occur during the anaphase, such as those driving transverse oscillations (12-14). Etc…”

There is therefore a clear discrepancy between the stochiometric model proposed here and this previously published work.

2) “We respectfully disagree with the premise of this question. While it is firmly established that LET-99 regulates the spatial distribution of forces (Krueger et. al, 2010), we believe that it is still unclear what precise properties of cortical force generators are altered by LET-99 and the exact spatial distribution of those properties influenced by LET-99 is also unclear.”

The title of the manuscript by (Krueger et al., 2010) is “LET-99 inhibits lateral posterior pulling forces during asymmetric spindle elongation in *C. elegans* embryos”.

Do not ignore this work, mention it in the text and discuss possible reasons for the observed discrepancies. In particular, the role of LET-99 is completely ignored in the stochiometric model.

Furthermore, the single centrosome experiment performed in (Krueger et al., 2010) (through zyg-1(RNAi), see Figure 4 and main conclusions : “After completing a mean of five transitions, the aster came to rest at a final position of 60% egg length, which is similar to the final midpoint of the spindle in wild-type embryos (Table 2).”) is in disagreement with results presented here (where a single centrosome, obtained after laser ablation of the other centrosome during metaphase, moved to the cell center).

3) “We added references in Table 1 to indicate the sources in the literature for the parameters we used in our simulations. The results of varying different parameters in the model is shown in Figure 5—figure supplement 1.”

As expected, the effect of microtubule catastrophe is huge in the stochiometric model. The authors used a catastrophe rate of 0.025/s, which seems extremely low and which they claim comes from (Kozlowski et al., 2007). However, I could not find this number anywhere is that manuscript. Rather, Kozlowski et al. used a cortical catastrophe rate in anaphase of 1 to 10/s. Check the reference to the parameter used in the paper and explain where the parameter comes from.

4) Write a little more on the link between partial correlations and correlations – this does not seem to be obvious.

---

## [Author Response]

The reviewers were all supportive of asking major revisions, described below. The feeling was that the authors need to improve the clarity and accuracy of this manuscript, especially the section related to the model, while shortening it during revision. Please pay special attention to point 9. Appropriate referencing and acknowledgements to others' work should be included.1) Some of the authors' conclusions disagree with some previously published findings and models (e.g. Astral microtubules generate pushing forces on spindle poles in Garzon-Corral et al., 2016), however the authors do not provide any potential explanation/hypothesis for this discrepancy. Could it be that the laser ablation experiments, which is the primary experimental base on which the authors draw their conclusions in this manuscript, have some intrinsic caveats (incomplete ablation of microtubules or side-effects of the ablation or rapid repolymerization of spindle microtubules or else) that would preclude drawing reliable conclusions? At minimum, it seems that some important controls are missing to draw strong conclusions (e.g. genetic evidence to back up their laser-ablation experiments and immunofluorescent staining of microtubules following ablation to demonstrate efficiency, etc). The only genetic control provided (i.e. spd-1(RNAi) compared to spindle microtubule ablation) is not really convincing as the two centrosomes clearly overshoot compared to controls following central spindle breakage, and the spindle over-elongates before going back to a control length (Figure 4I: the bump on the red curve). Would this be predicted by the stoichiometric model?

We believe there is no discrepancy between our results and previous published findings. The work of *Garzon-Corral et al., 2016,* is focused on forces in metaphase. Our study addresses forces on centrosomes at the end of anaphase. It is possible that different forces dominate at different times, which will be interesting and important to investigate in future studies.

Even with immunofluorescent staining or electron microscopy, it is not possible to conclusively prove that all microtubules are completely severed doing laser ablation. However, none of our conclusions would be altered if the laser ablation did not sever all the microtubules. This is because our conclusions are based on the direction of the motion the centrosome takes immediately after laser ablation. While incomplete severing or rapid microtubule regrowth after ablation might affect the extent of motion, it will not change the direction of motion. Furthermore, the regrowth can be directly observed on an approximately 30 seconds timescale, which occurs concordantly with the return of the centrosome to its original position (Figure 4—figure supplement 1 and 3).

The primary focus of this paper is the final position of centrosomes after the end of anaphase, when the centrosomes cease moving. Therefore, in this work we evaluated the Stoichiometric Model in the limit that the speed of centrosome motion is slow compared to polymerization and depolymerization dynamics of microtubules. This corresponds to using the quasi steadystate value of the microtubule length distribution and the P variables (the probability of microtubule attachment to a cortical force-generator) in Equation 10. If instead, we simulate centrosome motion using the complete theory (not restricted to the quasi steady-state limit), then the Stoichiometric Model does predict the observed overshoot of centrosome motion when spindle break in *spd-1(RNAi)* experiment (Author response image 1). While this effect is very interesting, it is not the subject of the current paper and we plan to address this and other aspects of spindle dynamics in a future manuscript.

**Author response image 1. sa2fig1:** Simulation of spindle elongation using the complete theory for control spindles (blue) and when the central spindle is removed at t=50 (red), corresponding to the spd-1(RNAi) experiment. Thus, the full Stoichiometric Model does predict an overshoot compared to controls following central spindle breakage, in which case the spindle over-elongates before going back to control length, as observed in experiments.

To further test the predictions of the Stoichiometric Model, we have also included comparisons between experiments and theory for *par-6 (RNAi)*. In both experiments and theory, *par6 (RNAi)* results in symmetric spindle positioning with an unperturbed final length. We described these results in the main text and added a new supplementary figure (Figure 5—figure supplement 5).

2) All the ablation experiments that are presented in this manuscript have been performed on the posterior half of the spindle. Since the stoichiometric model hypothesize that the two centrosomes are positioned by an identical mechanism (i.e. by a balance of cortical pulling forces), another important set of control experiments would be to ablate the different microtubule populations on the anterior side of the spindle.

To address this concern, we have added new experiments, ablating around the anterior centrosome with 7 plane-cuts, 16 cup-cuts parallel to spindle axis, and 11 cup-cuts perpendicular to the spindle axis (Figure 4—figure supplement 1 and 3). Our results indicate that the anterior centrosome is subject to net pulling forces in all directions similar to the posterior centrosome. We modified the main text to describe these results.

3) The stoichiometric model posits that centrosomes are positioned by a balance of pulling forces acting in different directions. This is not consistent with genetic evidence on the role of LET-99, which, in the posterior half of the embryo, restricts cortical pulling forces to the posterior-most region of the cortex. How do the authors reconcile this discrepancy between the design of their mathematical model and this genetic data?

We respectfully disagree with the premise of this question. While it is firmly established that LET-99 regulates the spatial distribution of forces (Krueger et. al, 2010), we believe that it is still unclear what precise properties of cortical force generators are altered by LET-99 and the exact spatial distribution of those properties influenced by LET-99 is also unclear. Furthermore, our experimental results demonstrate that the anterior and posterior centrosomes are subject to pulling forces directed both parallel and perpendicular to the spindle axis. Thus, cortical pulling forces are not restricted to the posterior most region of the cortex. Since our simulations explicitly account for the discrete location and behaviors of individual cortical force generators, it would be straightforward to modify them to investigate the consequences of different possible models of LET-99 activity. While that is a very interesting direction for future work, such effects are not necessary to describe any of the experiments we present in this manuscript, so we did not incorporate these additional complications into our simulations.

4) Another important parameter in the stoichiometric model is microtubule dynamics. In particular, the catastrophe frequency of astral microtubules is likely to have a major effect on the outcome of the simulation. The parameters (growth rate, catastrophe frequency and nucleation rate) used in the simulation are listed in a table at the end of the manuscript but it is not clear how these values have been set.

We added references in Table 1 to indicate the sources in the literature for the parameters we used in our simulations. The results of varying different parameters in the model is shown in Figure 5—figure supplement 1.

5) A model of spindle positioning in *C. elegans* that relies solely on cortical pulling forces, and which the authors omitted to cite in their current manuscript, has already been published (Bouvrais et al., 2018). This previous model has the advantage that it fully accounts for the effect of the LET-99 loss-of-function observed in vivo on spindle positioning and elongation. Does the model in the current manuscript represents a significant advance in our understanding of spindle elongation and positioning?

The model in the current manuscript is very different from the model of Bouvrais et al., 2018, on both a conceptual and technical level. From the conceptual perspective, the model of Bouvrais et al. includes both a centering spring and separate destabilizing pulling forces. In their model, the final position of centrosomes results from a balance of the spring force, which favors a shorter more centered spindle, and the pulling forces, which favor a longer asymmetrically positioned spindle. In previous work it has been argued that the centering spring force results from microtubule pushing (Pecreaux et al., 2016). In contrast, the Stoichiometric Model only contains pulling forces without the need to invoke a separate centering spring force. Thus, in the Stoichiometric Model, both spindle elongation and asymmetric positioning result only from pulling forces.

On a technical level, the model of Bouvrais et al. is also very different from the present model. Bouvrais et al. write down phenomenological equations justified by intuitive arguments. In contrast, the Stoichiometric Model presented here, is a constructive model derived from the fundamental equations of dynamic instability of microtubules, the biochemistry of molecular motors, and explicitly accounts for the geometry and mechanics of the interactions of microtubules and force-generators. So, yes, we consider this new model a very substantial advance in the understanding of spindle positioning and elongation.

6) The fact that gpr-1/2 and par-2 are important regulators of spindle elongation during anaphase, and thus of final spindle length has been demonstrated many years ago. Therefore I'm not convinced that the QTL analysis performed in this study really highlights a novel feature of the mechanism that controls spindle elongation in the one-cell *C. elegans* embryo.

We respectfully disagree with the reviewer’s comment.

While gpr-1/2 and par-2 are well known to be involved in spindle elongation, we are unaware of previous works demonstrating par-2’s role in final spindle length. The connection between spindle elongation and final spindle length is not obvious. For example, *spd-1(RNAi)* dramatically increases the rate of spindle elongation, but has no impact on final spindle length (Figure 4I). Furthermore, while par-6 is also involved in spindle elongation, *par-6(RNAi)* also does not affect final spindle length (Figure 5—figure supplement 5).

We would like to emphasize that the use of recombinant inbred lines provides a novel and rigorous means of testing different classes of models by investigating correlations and partial correlations between traits. Using sequencing data to identify genetic factors that influence those traits is an additional benefit. Such forward genetic screens allows novel factors to be identified if they are present. Since the QTLs we identified are associated with the genes previously known to impact the spindle, this suggest that the “parts list” of proteins involved in the spindle might be nearly complete. Because identified QTLs are based on natural genetic variations, they also can provide insight into the evolutionary genetics of the spindle.

7) They carry out some RNAi experiments of the gpr1 and par mutants to examine spindle length. Under these conditions, one loses the cortical interaction to the microtubule. What does their model predict in the absence of cortical forces in terms of spindle length?

Our model predicts that the spindle does not elongate in the absence of pulling forces. We speculate that the *gpr-1/2(RNAi)* and *par-2(RNAi)* experiments produce only partial knockdowns, and hence pulling forces are reduced, but still present.

8) The authors do not give details about their stochastic simulation. I am still not sure what they really did. Did they simulate microtubules growing out of centrosomes and then calculate the forces to obtain the centrosome dynamics? Or did they use the probabilities for microtubule attachment to a force generator and based their simulation on these probabilities?

We have improved the presentation in the Materials and methods section to explain more precisely how the simulations were performed. We have added a new section titled “Simulation procedure” to the Materials and methods. In our simulations, we explicitly account for the location and orientation of each cortical force-generator, as well as the position of the two centrosomes. At each time step of the simulations, we calculate the force exerted by each force-generator on the centrosomes using Equation 6 for the impingement rate of microtubules on that force-generator and Equation 11 to account for the stoichiometric interactions through the probability of attachment. We then use Equation 13 to calculate the net pulling force on each centrosome, and Equation 14 to update the position of centrosomes accounting for both drag on the centrosomes and central spindle viscosity.

9) The authors did not do a good job in explaining the mechanism by which the stoichiometric model leads to centrosome centering. It is still true that the probability of a force generator binding a microtubule increases with the centrosome coming closer to the force generator. The authors state that "In this model, the stable positioning of centrosomes results from the stoichiometric interaction between MTs and CFGs, which prevents the destabilizing feedback present in previous models of cortical pulling forces.". This did not help to understand, why the model works. It seems that the central ingredient is that the forces are along the direction of the microtubules and that these forces vary in magnitude. This is because the force generated by the force generator is perpendicular to the cell boundary and it is this force that is the same in magnitude for all force generators; the component projected into the direction of the microtubule is not. As the centrosome moves to or away from the boundary, the angle of the microtubules with the boundary and thus the force they experience changes. If I understood correctly, this is why the centrosome position is stabilized: the closer you get to the boundary the smaller the component in the direction of the microtubule typically becomes (the simplest situation to see this mechanism at work is a centrosome between two parallel plates, where there are two force generators on each plate). The condition of having at most one microtubule bound to a force generator prevents the binding of more and more microtubules to the force generator nearest to the centrosome, which would lead to an instability of the central centrosome position. – Independently of whether the mechanism I invoke here is correct or not, I would urge the authors to explain the mechanism of stabilizing the central centrosome position and not just to state that their mechanism does so.

We agree that we neglected to provide a complete explanation of the stabilizing mechanisms of our model. The reviewer has accurately described the two crucial aspects. There is a loss of pulling force due to the increasing obliqueness of applied pulling forces to the centrosome as it approaches the cortex. Since each force generator can bind only one microtubule (this is stoichiometry), this geometric effect is not compensated for by the proximity of the centrosome providing yet more microtubules to bind. These combined effects are indeed central to the functioning of the model, and to emphasize this point we have now provided a new simulation of a centrosome between two parallel plates as suggested. We used this to produce a new figure (Figure 5E) that clearly illustrates the mechanism of stable centrosome positioning. We have also added an explanation of this in the main text.

[Editors' note: further revisions were suggested prior to acceptance, as described below.]

The reviewers agreed that the manuscript was improved, yet a number of points raised by the reviewers have not been addressed in the text. Please go through each point (in previous decision letter) and rather than only respond to the reviewers, use their comments to guide the editing of your manuscript to improve clarity of the text and mention the changes to the text in the rebuttal letter, shortening the text where appropriate to streamline. In the title, "interactions" is plural ut the verb is singular, this needs correcting.

We corrected the title.

2) “We respectfully disagree with the premise of this question. While it is firmly established that LET-99 regulates the spatial distribution of forces (Krueger et. al, 2010), we believe that it is still unclear what precise properties of cortical force generators are altered by LET-99 and the exact spatial distribution of those properties influenced by LET-99 is also unclear.”The title of the manuscript by (Krueger et al., 2010) is “LET-99 inhibits lateral posterior pulling forces during asymmetric spindle elongation in *C. elegans* embryos”.Do not ignore this work, mention it in the text and discuss possible reasons for the observed discrepancies. In particular, the role of LET-99 is completely ignored in the stochiometric model.

We believe there is no discrepancy between our study and (Krueger et al., 2010). While in our study we only considered a “two-domain” CFGs model for simplicity, it is straight forward to extend the Stoichiometric Model to “three-domain” CFGs model. We added a paragraph to specifically refer to this publication.

3) “We added references in Table 1 to indicate the sources in the literature for the parameters we used in our simulations. The results of varying different parameters in the model is shown in Figure 5—figure supplement 1.”As expected, the effect of microtubule catastrophe is huge in the stochiometric model. The authors used a catastrophe rate of 0.025/s, which seems extremely low and which they claim comes from (Kozlowski et al., 2007). However, I could not find this number anywhere is that manuscript. Rather, Kozlowski et al. used a cortical catastrophe rate in anaphase of 1 to 10/s. Check the reference to the parameter used in the paper and explain where the parameter comes from.

In (Kozlowski et al., 2007), they set the cytoplasmic catastrophe rate to 0.01/s (row 12 in table on page 504), which is quite similar to the value that we use, 0.025/s. In the Stoichiometric Model, microtubules undergo catastrophe after detaching from force-generators, so the detachment rate, κ, can also be thought of as the catastrophe rate of attached microtubules. We used a value of κ of 0.1 s^-1^, which is smaller than the value used by (Kozlowski et al., 2007), but in a range consistent with measurements from (Redemann et al., 2010). We also investigated the impact of varying these parameters (Figure 5—figure supplement 1). We have added a sentence explaining that κ can be thought of as the catastrophe rate of microtubules attached to force generators.

4) Write a little more on the link between partial correlations and correlations – this does not seem to be obvious.

We added more explanation for analysis of correlation and partial correlation in Materials and methods.